# A simple, efficient, and scalable method to generate oocyte-like cells in vitro

Ami Banno[1],*, Kana Mizuno[1],*, Mizuki Sakamoto[1], Sota Komatsubara[1], Kenjiro Shirane[2], Katsuhiko Hayashi[2,3], Nobuhiko Hamazaki[4,5,6,7,8], Koshi Imami[9], Shigenobu Yonemura[10,11], Takashi Ishiuchi[1]

Understanding the molecular basis of oocyte identity and function is essential not only for basic biology but also for clinical applications, as it is closely linked to female infertility. However, technical challenges remain in advancing this understanding, mainly because of the difficulty in obtaining a sufficient number of oocytes. In this study, through refining previously reported three-dimensional protocols, we established a two-dimensional culture method that efficiently generates oocyte-like cells, referred to as mini-oocytes, from mouse embryonic stem cells. This method requires minimal labor, does not rely on supporting somatic cells, and leverages a transcription factor–mediated approach for oocyte-like cell generation. Our transcriptome and proteome analyses revealed significant similarities between in vitro–derived mini-oocytes and in vivo oocytes, despite their relatively smaller size. Furthermore, we demonstrated the utility of mini-oocytes for investigating oocyte-specific molecular features through a small-scale knockout screen targeting the subcortical maternal complex. Given the simplicity, efficiency, and scalability of the mini-oocyte induction method, it offers a practical platform for conducting experiments that are otherwise challenging with in vivo oocytes.

## Introduction

Oocyte development involves the formation of primordial germ cells (PGCs), entry into meiosis, meiotic arrest at the diplotene stage of meiotic prophase I, primordial follicle formation, and subsequent oocyte growth, ultimately resulting in preovulatory fully grown oocytes (FGOs) within the ovary (Richards & Pangas, 2010; Pepling, 2012; Saitou et al, 2012; Suzuki et al, 2015). During oocyte growth, cell size increases and maternal RNA and proteins accumulate (Pan et al, 2005; Vastenhouw et al, 2019; Wu & Dean, 2020). FGOs resume meiosis upon hormonal stimulation and are ovulated as metaphase II (MII) oocytes. MII oocytes, upon fertilization by sperm, give rise to zygotes, which serve as the foundation of the next generation.

Understanding the molecular basis of oocyte development is critically important, as it ensures the continuity of life. Genetic studies have identified genes necessary for oocyte development (Jagarlamudi & Rajkovic, 2012; Sanchez & Smitz, 2012). Based on these understandings, in vitro reconstitution of the entire process of mouse female germ cell development has been achieved and delineated the extrinsic signals necessary and sufficient for the oocyte development starting from mouse embryonic stem cells (mESCs) or its counterpart, naïve epiblast (Hikabe et al, 2016; Saitou & Hayashi, 2021). This also enabled the generation of oocytes from induced pluripotent stem (iPS) cells, offering a potential solution to female infertility, such as that caused by the absence of oocytes. In another study, on the basis of the identification of transcription factors (TFs) specifically expressed in oocytes, oocyte-like cells were efficiently generated in vitro directly from mESCs through the forced expression of a set of TFs (Hamazaki et al, 2021). This work indicated that the co-expression of eight transcription factors (8Fs), Nobox, Figla, Tbpl2, Lhx8, Stat3, Dynll1, Sub1, and Sohlh1, can convert mESCs to oocyte-like cells. Among them, four factors (4Fs), Nobox, Figla, Tbpl2, and Lhx8, were found to be the minimum cocktail sufficient to induce oocyte-like cells. It has been confirmed that oogenesis is separable from meiosis (Dokshin et al, 2013). Indeed, the TFs directly activated the oocyte-specific transcriptional network and generated oocyte-like cells without initiating the meiotic program.

Thus, to date, the generation of oocytes (or oocyte-like cells) has been achieved in vitro either through the reconstitution of the in vivo developmental process or through the forced expression of TFs. The prominent difference between these two methods is the

---

[1]Faculty of Life and Environmental Sciences, University of Yamanashi, Yamanashi, Japan  [2]Department of Genome Biology, Graduate School of Medicine, The University of Osaka, Osaka, Japan  [3]Premium Research Institute for Human Metaverse Medicine (WPI-PRIMe), The University of Osaka, Osaka, Japan  [4]Departments of Obstetrics and Gynecology, University of Washington, Seattle, WA, USA  [5]Department of Genome Sciences, University of Washington, Seattle, WA, USA  [6]Institute for Stem Cell and Regenerative Medicine, University of Washington, Seattle, WA, USA  [7]Brotman Baty Institute, Seattle, WA, USA  [8]Seattle Hub for Synthetic Biology, Seattle, WA, USA  [9]Proteome Homeostasis Research Unit, RIKEN Center for Integrative Medical Sciences, Yokohama, Japan  [10]Laboratory for Ultrastructural Research, RIKEN Center for Biosystems Dynamics Research, Kobe, Japan  [11]Department of Cell Biology, Tokushima University Graduate School of Medicine, Tokushima, Japan

Correspondence: tishiuchi@yamanashi.ac.jp
*Ami Banno and Kana Mizuno contributed equally to this work

 

culture condition for the induction. As the former involves the mimic of the intricate developmental process from the naïve epiblast, sequential changes of medium components that can generate specific cell types, such as epiblast-like cells, PGC-like cells, and oogonia, are necessary (Hayashi et al, 2017). In contrast, as the latter does not involve the sequential generation of these cell types, the culture condition can be simplified, and only a single culture condition is required for the differentiation into oocyte-like cells (Hamazaki et al, 2021). After these inductions, both methods required the three-dimensional culture of reconstituted ovaries on membranes and isolation and culture of each follicle for maturation.

Despite recent advances in germ cell research, several technical limitations remain. For instance, performing biochemical assays or genetic screening using oocytes is challenging because of the difficulty in obtaining a sufficient number of oocytes. As a result, a comprehensive understanding of the molecular networks that govern oocyte development and function remains incomplete. In this study, prompted by the relative simplicity of inducing oocyte-like cells using specific TFs, we speculated that modifying or adjusting the induction method might be useful for obtaining sufficient oocyte material to conduct such assays. Therefore, we refined the culture conditions to develop a more simplified two-dimensional culture system that can efficiently generate oocyte-like cells with minimal labor. Our analyses revealed their significant similarity to in vivo oocytes despite their smaller size, as well as the presence of a molecular network unique to oocytes, thereby offering a practical alternative platform for oocyte research.

# Results

### Generation of oocyte-like cells by a simple two-dimensional culture

Previous studies employed three-dimensional culture for the induction of oocyte-like cells, which involves generating cell aggregates and transferring each of them onto porous membranes (Hikabe et al, 2016; Hamazaki et al, 2021). In addition, as gonadal somatic cells have been used to reconstruct ovarian structures, recovery of oocyte-like cells required sophisticated techniques (Hayashi et al, 2017). These steps are highly labor-intensive, making the experiment extremely challenging when a large number of oocyte-like cells (e.g., >10$^5$), particularly well-grown oocytes, are required. Therefore, to enable the efficient recovery of oocyte-like cells in vitro, we established a culture condition that generates cells with oocyte-like morphology in a two-dimensional system based on the TF-mediated induction method. For our assays, we used mESC lines in which the expression of 4Fs (Nobox, Figla, Tbpl2, and Lhx8) or 8Fs (the 4Fs plus Stat3, Dynll1, Sub1, and Sohlh1) can be induced in the presence of Shield-1, including the one used in a previous study (Hamazaki et al, 2021). In the Shield-1 system, transcription factors are fused to a destabilization domain and actively degraded in the absence of Shield-1, whereas the addition of Shield-1 stabilizes these fusion proteins, thereby enabling immediate and sustained transcription factor expression. For

induction, a StemPro34-based medium containing Shield-1, stem cell factor (SCF), and Y-27632 (a ROCK inhibitor) was used, as this condition efficiently supported oocyte growth before (Hamazaki et al, 2021). In the subsequent experiments, we exchanged the medium every 3 d using the same culture medium, except that Shield-1 was omitted starting from day 12 (Fig 1A). After plating the cells in a standard culture plate on day 0, cells changed their morphology and gradually increased cell size showing round shape (Fig 1B and C). Although significant cell death was observed during a few days after the induction, a fraction of the cells consistently survived. On average, 13.9% and 46.7% of the plated 4F- and 8F-mESCs eventually survived, respectively (Fig 1D), and virtually almost all of the survived cells were induced into oocyte-like cells (Fig 1B). In addition to the survival rate, we noted some differences between 4F-mESCs and 8F-mESCs; 8F-mESCs increased cell size faster than 4F-mESCs (Fig 1C). In addition, although 4F-mESCs exhibited a flat cell morphology reminiscent of highly differentiated cells on days 1 to 3, 8F-mESCs still indicated dome-like colonies similar to mESC colonies (Fig 1B). The reason and significance of these differences are currently unclear. Both of them increased in size until day 21 (Fig 1C); however, we observed a cessation of cell size increase and collapse in cell morphology during culture beyond day 21 (Fig S1A). Therefore, we determined the maximum induction period to be 21 d. On day 21, the average major axis of oocyte-like cells generated from 4F-mESCs and 8F-mESCs was 34.6 and 39.8 $\mu m$, respectively. As their size was smaller than that of in vivo FGOs (with a diameter of 70–80 $\mu m$), we refer to these oocyte-like cells as "mini-oocytes" hereafter (Fig 1E). During the mini-oocyte induction, changing the medium every 3 d only is required, highlighting the simplicity of this method (Fig 1A). To assess the purity of mini-oocytes after induction, we examined the expression of oocyte growth markers. At day 21, 98.5% of cells derived from 4F-mESCs were positive for ZP2 staining (the presence of a zona pellucida in mini-oocytes is described later), and 92.2% of cells derived from 8F-mESCs, carrying a *Npm2-mCherry* reporter cassette (Hamazaki et al, 2021), were positive for Npm2-mCherry reporter expression, further indicating the high efficiency of mini-oocyte induction (Figs 1F and S1B). In addition, the availability of mini-oocytes can be easily adjusted by varying the number of plating mESCs and the size of cell culture plates or dishes, indicating its high scalability (we were able to obtain more than 10$^6$ mini-oocytes from a single experiment). Thus, this mini-oocyte induction system offers several advantages, including a simple procedure, no requirement for gonadal somatic cells, and an almost unlimited supply of oocyte-like cells, in contrast to previous oocyte induction systems.

### Requirement of culture supplements for the generation of mini-oocytes

We next examined the requirement of culture supplements, Shield-1, SCF, and Y-27632, for the generation of mini-oocytes using 4F-mESCs. To this end, we removed each of them from the beginning of the induction. Without Shield-1, mESCs continued to proliferate while displaying differentiated morphology, whereas the presence of Shield-1 rapidly halted cell proliferation (Fig S1C). Thus, TF-mediated activation of the oocyte transcription network is

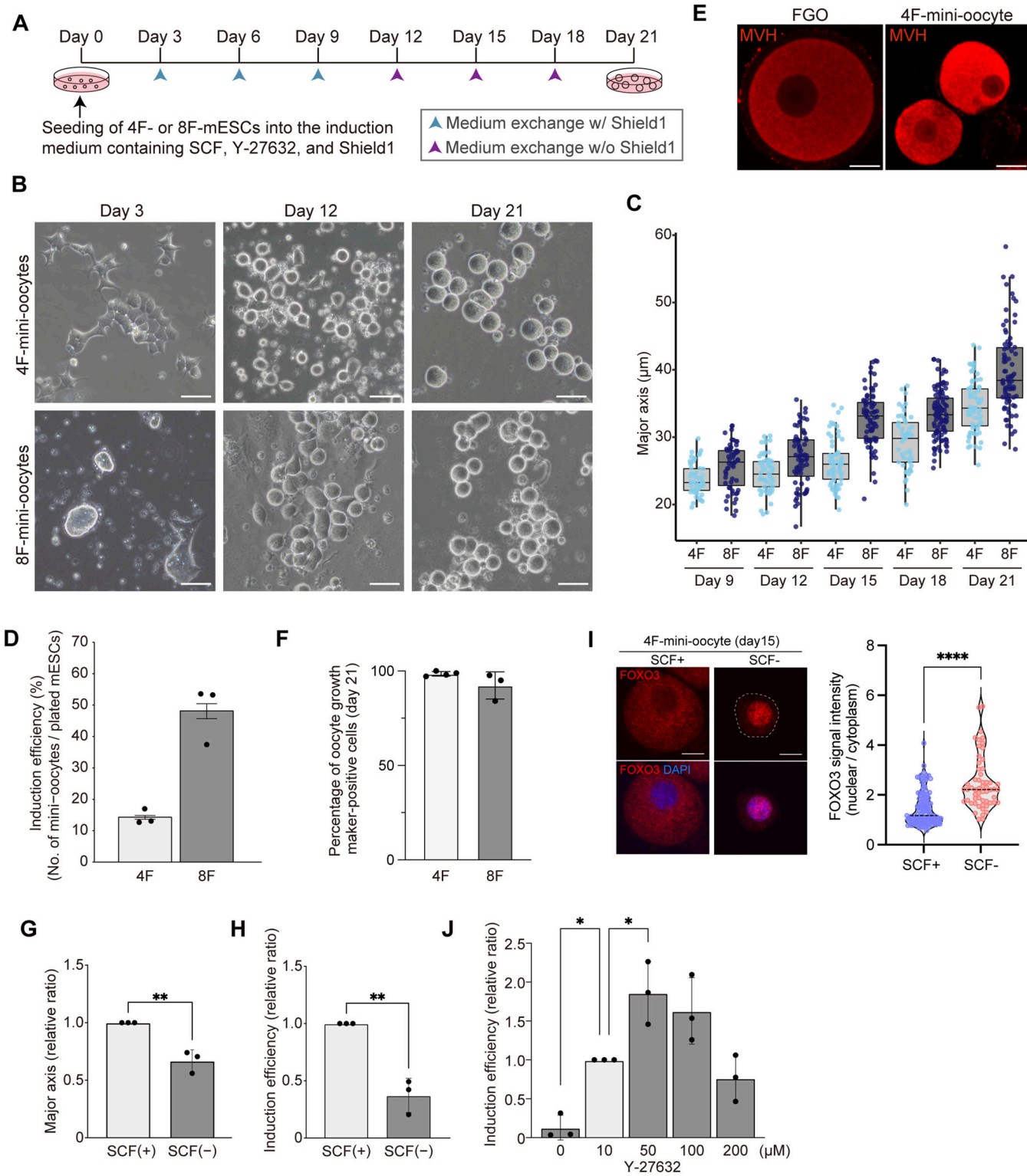

**Figure 1. Establishment of the mini-oocyte induction method.**

**(A)** Overview of the mini-oocyte induction method. **(B)** Representative bright-field images showing the morphology of mini-oocytes during induction. Scale bar, 50 μm. **(C)** Boxplots showing the increase in the major axis of mini-oocytes during induction. **(D)** Bar plots showing the induction efficiency of mini-oocytes using 4F- or 8F-mESCs. The induction efficiency represents the ratio between the number of total mini-oocytes and that of plated mESCs. **(E)** Immunofluorescence images showing the differential size of in vivo FGOs and mini-oocytes. Anti-MVH antibody was used for the immunofluorescence. Scale bar, 20 μm. **(F)** Bar plots showing the percentage of growing oocyte marker-positive cells upon mini-oocyte induction. A proportion of ZP2 and Npm2-mCherry reporter-positive cells were examined for 4F- and 8F-mESCs, respectively, at day 21 of induction. **(G)** Bar plots showing the relative ratio of the major axis in the presence (+) or absence (−) of SCF. **(H)** Bar plots showing the relative ratio of induction efficiency under the presence (+) or absence (−) of SCF. **(I)** Left, immunofluorescence images of FOXO3 staining in day 15 4F-mini-oocytes under the

sufficient to halt cell proliferation, whereas it remains unclear how cell cycle arrest occurs in the absence of meiosis. In contrast, when SCF was omitted, cell proliferation ceased, and the cells acquired a round shape, similarly as observed in a normal condition. However, they were not able to increase cell size, and cell viability was reduced (Figs 1G and H and S1D). Therefore, although SCF appears to be dispensable for the initial acquisition of oocyte-like features, it is required for their survival and growth. This is consistent with previous observations that mutant mice lacking functional KIT, the SCF receptor, exhibit impaired oocyte growth while preserving primordial follicle oocytes (Yoshida et al, 1997; Saatcioglu et al, 2016). Consistent with the in vivo SCF-KIT signaling pathway, which promotes FOXO3 relocalization from the nucleus to the cytoplasm to initiate oocyte growth (Liu et al, 2006), FOXO3 tended to be excluded from the nucleus in a SCF-dependent manner in mini-oocytes (Fig 1I). We also examined the role of Y-27632, a Rho-kinase inhibitor (10 $\mu$M was used as a standard concentration). As Y-27632 is generally used to suppress cell death, we tested its effect on cell viability by altering its concentration. As expected, the removal of Y-27632 decreased cell viability. In contrast, increasing its concentration to 50 $\mu$M was most effective in suppressing cell death and, consequently, increased the number of available mini-oocytes (Fig 1J). Taken together, these results indicate that each culture supplement plays a distinct role in the efficient generation of mini-oocytes.

### The transcriptome of mini-oocytes exhibits high similarity to that of in vivo oocytes

To determine the characteristics of mini-oocytes, we performed RNA-sequencing (RNA-seq) analyses. RNA-seq data from in vivo differentiating oocytes were used to identify at which stage of oocytes mini-oocytes most closely resemble. Principal component analysis (PCA) revealed that mini-oocytes generated from 4F-mESCs (4F-mini-oocytes) and those from 8F-mESCs (8F-mini-oocytes) on day 21 exhibit highly similar transcriptome profiles (Fig 2A and B and Table S1). This suggests that their difference in cell size is largely independent of their transcriptome profile. Furthermore, comparison with other RNA-seq datasets revealed that mini-oocytes at day 21 possess a transcriptome profile that is intermediate but leaning more toward FGOs than secondary oocytes, possibly reflecting their smaller size compared with FGOs (Fig 2A). Thus, mini-oocytes at day 21 can be considered partially FGOs, at least at the transcriptome level. To investigate the transition process from mESCs to mini-oocytes at day 21, we also performed RNA-seq on 4F-mini-oocytes at day 7, when their round morphology becomes apparent. In the PCA plots, 4F-mini-oocytes at day 7 were positioned between mESCs and day 21 mini-oocytes, indicating an intermediate state during their transition (Fig 2A). In addition, along the PC1, mini-oocytes at day 7 located close to in vivo postnatal day 4 small and large oocytes, suggesting that mini-oocytes at day 7 might have a part of the characteristics of primordial and primary oocytes. To gain insights into the mini-

oocyte induction process, we examined the expression of marker genes. As expected, PGC, early germ cell, and meiosis markers were not induced during the mini-oocyte induction (Fig S2). In contrast, the up-regulation of growing oocyte markers, including *Zp1*, *Zp2*, *Zp3*, *Gdf9*, and *Npm2*, was evident (Fig 2C). Thus, these results indicate that an oogenesis-like process proceeds during mini-oocyte induction.

Scatter plots of gene expression levels indicated a high degree of linearity between mini-oocytes at day 21 and in vivo FGOs, in contrast to the comparison between mini-oocytes at day 21 and mESCs (Figs 2D and S3A). This suggests that although the expression amplitude of the expressed genes moderately differs between mini-oocytes at day 21 and in vivo FGOs, the expressed gene set is largely shared. To understand the difference in gene expression, we extracted differentially expressed genes (DEGs) between them (FDR < 0.05). Thousands of up-regulated and down-regulated genes were detected, as expected from the observed difference in PCA (Fig 2E). The up-regulated and down-regulated genes observed in 4F-mini-oocytes and 8F-mini-oocytes, compared with in vivo FGOs, showed substantial overlap (Figs 2F and S3B). Gene ontology analysis revealed that genes associated with translation and mitochondrial activity are up-regulated in mini-oocytes, suggesting that mini-oocytes are more active in protein synthesis and energy production compared with in vivo FGOs (Fig 2G). KEGG pathway analysis indicated that genes involved in the cAMP signaling pathway are down-regulated in mini-oocytes (Fig S3C and see the Discussion section). Overall, these results indicate that although mini-oocytes are not identical to in vivo oocytes, they exhibit transcriptome profiles that resemble those of in vivo oocytes.

### Activation of oocyte-specific transposable elements in mini-oocytes

In addition to genes, specific transposable elements (TEs) are activated in a cell type–dependent manner. Therefore, we examined whether oocyte-specific TEs are activated in mini-oocytes. A comparison between in vivo FGOs and mESCs revealed higher expression of members of the MT subfamily (MTA-C), which belong to the mammalian apparent LTR retrotransposons, as well as the RLTR10 elements in FGOs (Fig 3A), consistently with the previous observation (Brind'Amour et al, 2018; Evsikov et al, 2006; Peaston et al, 2004). Interestingly, 4F- and 8F-mini-oocytes at day 21 also exhibited high expression levels of the MT and RLTR10 elements (Fig 3A). As a result, the retrotransposon expression profiles of mini-oocytes were highly similar to those of FGOs (Fig 3B).

Retrotransposons can function as alternative promoters and generate chimeric transcripts. Our observations thus suggest the presence of oocyte-specific chimeric transcripts in mini-oocytes. One of the most well-characterized oocyte-specific chimeric transcripts is a truncated isoform of *Dicer1* (*Dicer1^O*), which is responsible for active RNAi in mouse oocytes (Flemr et al, 2013). In this case, a solo MTC LTR located within the intron 6 of the

---

presence (+) or absence (−) of SCF. Scale bar, 10 $\mu$m. Right, violin plots showing the nuclear and cytoplasmic FOXO3 signal intensity ratio. **(J)** Bar plots showing the relative ratio of induction efficiency under the different concentration of Y-27632.

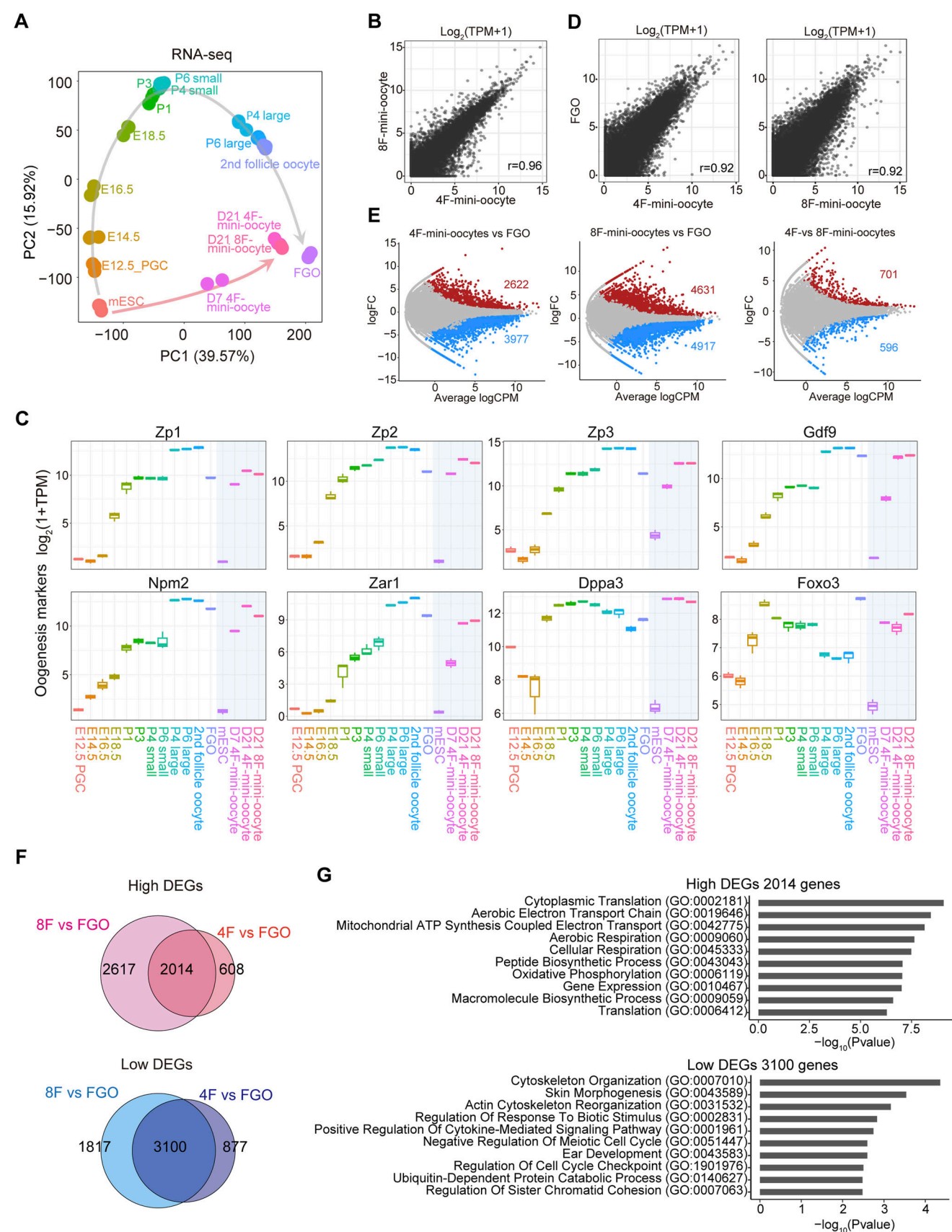

*Dicer1* gene acts as an alternative promoter to produce the shorter isoform. Notably, careful inspection of RNA-seq data revealed reads spanning from the MTC element to the canonical exon 7 in 4F- and 8F-mini-oocytes, as well as FGOs, indicating MTC-driven transcription of *Dicer1^O^* (Fig 3C). In addition to *Dicer1^O^*, we also detected oocyte-specific LTR-driven expression of genes such as *Piwil1*, *Dnmt3b*, *Th*, and *Bmp5* (Brind'Amour et al, 2018) (Fig S3D). These findings strongly suggest that mini-oocytes can serve as a useful model to study the roles of oocyte-specific retro-transposons and chimeric transcripts.

### Mini-oocytes exhibit proteomic profiles similar to in vivo oocytes

RNA-seq data indicated that transcripts present in in vivo oocytes are also expressed in mini-oocytes. To further investigate whether these transcripts are indeed translated into proteins, we performed proteome analysis on in vivo FGOs and 4F- and 8F-mini-oocytes at day 21 (Table S2). Triplicate samples were prepared for each group, and proteins detected in all three replicates were considered to be faithfully expressed. Among the 3,868 proteins detected in the in vivo FGO sample, 3,568 (92%) and 3,429 (89%) were commonly identified in the 4F- and 8F-mini-oocyte samples, respectively (Fig 4A). Thus, most of the proteins that are relatively abundant and detectable by proteome analysis in FGOs are also expressed in mini-oocytes. However, some proteins appeared to be excessively expressed in mini-oocytes (Fig 4A and B). This apparent elevation may stem from the larger input amount of the mini-oocyte samples compared with the FGO samples (see the Materials and Methods section). Nevertheless, the higher expression of these proteins was consistent with RNA-seq data (Figs 4B and C and S3E), and gene ontology analysis of these excessive proteins in mini-oocytes revealed that they are related to translation, such as ribosome biogenesis, as observed in the transcriptome analysis (Fig 4D). We speculate that these differences in proteomic profiles between mini-oocytes and in vivo oocytes may reflect the culture environment, which substantially differs from its in vivo counterpart. In addition, we observed that proteins related to cholesterol transport are detected in FGOs but not in mini-oocytes, suggesting the lack or down-regulation of these proteins in mini-oocytes (Fig 4E). Collectively, these results indicate that although there are some differences between mini-oocytes and FGOs, many of the proteins expressed in in vivo FGOs are also present in mini-oocytes.

### Zona pellucida–like structures are formed in mini-oocytes

RNA-seq and proteome analyses indicated a similarity between mini-oocytes and in vivo FGOs, and Zp1, Zp2, and Zp3, the zona pellucida components, were expressed in mini-oocytes at the RNA and protein levels (Fig 5A). However, the zona pellucida was not visible under an optical microscope. Thus, to further investigate the characteristics of mini-oocytes, we performed electron microscopy (EM) on mini-oocytes at day 21, in vivo FGOs, and mESCs (Fig 5B). This analysis revealed the presence of a zona pellucida–like structure surrounding the plasma membrane of mini-oocytes, although it was thinner and fainter than that of in vivo FGOs. Consistently with the EM observation, immunofluorescence staining detected ZP2 and ZP3 signals on mini-oocytes (Fig 5C and D). In addition to the zona pellucida, we observed well-developed microvilli upon mini-oocyte induction, a characteristic feature of in vivo oocytes (Figs 5B and S4). In contrast, cytoplasmic lattices, a unique structure found in oocytes (Wassarman & Josefowicz, 1978), were not detectable in mini-oocytes, although we cannot exclude their presence in a subset of cells (Fig S4). In addition, we observed differences in mitochondrial morphology between mini-oocytes and FGOs, possibly reflecting the results from transcriptome analyses. Thus, mini-oocytes partially recapitulate structural characteristics of in vivo oocytes, suggesting the necessity of further improvements to better recapitulate the in vivo oocytes at the ultrastructure level.

### Mini-oocytes develop molecular networks reminiscent of in vivo oocytes

We next investigated whether mini-oocytes could serve as an in vitro model for understanding the molecular networks unique to oocytes. Oocytes are characterized by the expression of oocyte-specific genes. Based on RNA-seq and proteome data, we confirmed the expression of such genes required for oogenesis, fertilization, maternal RNA regulation, DNA methylation (imprinting), and components of the subcortical maternal complex (SCMC) (Figs 5A and 6A).

DNA methylation is uniquely regulated in oocytes (Sendzikaite & Kelsey, 2019; Xia & Xie, 2020). Notably, UHRF1 and DNMT1, which are typically localized in the nucleus, are excluded from it and predominantly found in the cytoplasm of oocytes (Hirasawa et al, 2008; Maenohara et al, 2017). It has been reported that DPPA3 (also known as STELLA) is responsible for the cytoplasmic localization of UHRF1. Through this mechanism, DPPA3 protects oocytes from

---

**Figure 2. Transcriptome of mini-oocytes exhibits high similarity to that of in vivo oocytes.**
**(A)** Plots showing the principal component analysis of transcriptome data. RNA-seq data for mouse embryonic stem cells, in vivo female germ cells at E12.5, E14.5, E16.5, E18.5, postnatal day 1 (P1), P3, P4, P6, and secondary follicle stage, and fully grown oocytes (FGOs) were used for comparison. Published RNA-seq data for FGOs (GSE183969) or differentiating oocytes (GSE79729 and GSE128305) were used to analyze their transcriptional profiles (Hikabe et al, 2016; Shimamoto et al, 2019; Yano et al, 2022). **(B)** Scatter plots showing the correlation of gene expression between 4F-mini-oocytes and 8F-mini-oocytes. Gene expression levels are shown by $log_2$(TPM + 1). Pearson's correlation coefficient (r) is also indicated. **(A, C)** Box plots showing the expression of indicated genes related to oogenesis. RNA-seq data indicated in (A) were used. **(D)** Scatter plots showing the correlation of gene expression between FGOs and 4F-mini-oocytes or 8F-mini-oocytes. Gene expression levels are shown by $log_2$(TPM + 1). Pearson's correlation coefficient (r) is also indicated. **(E)** MA plots showing the differential gene expression among FGOs, 4F-mini-oocytes, and 8F-mini-oocytes. FDR < 0.05 was considered as differentially expressed. **(F)** Venn diagrams showing the overlap of differentially expressed genes. Top, overlap of genes highly expressed in mini-oocytes compared with FGOs. Bottom, overlap of genes lowly expressed in mini-oocytes compared with FGOs. **(G)** Gene ontology analysis of differentially expressed genes. Top, genes highly expressed in both 4F- and 8F-mini-oocytes in comparison with FGOs were analyzed. Bottom, genes lowly expressed in both 4F- and 8F-mini-oocytes in comparison with FGOs were analyzed.

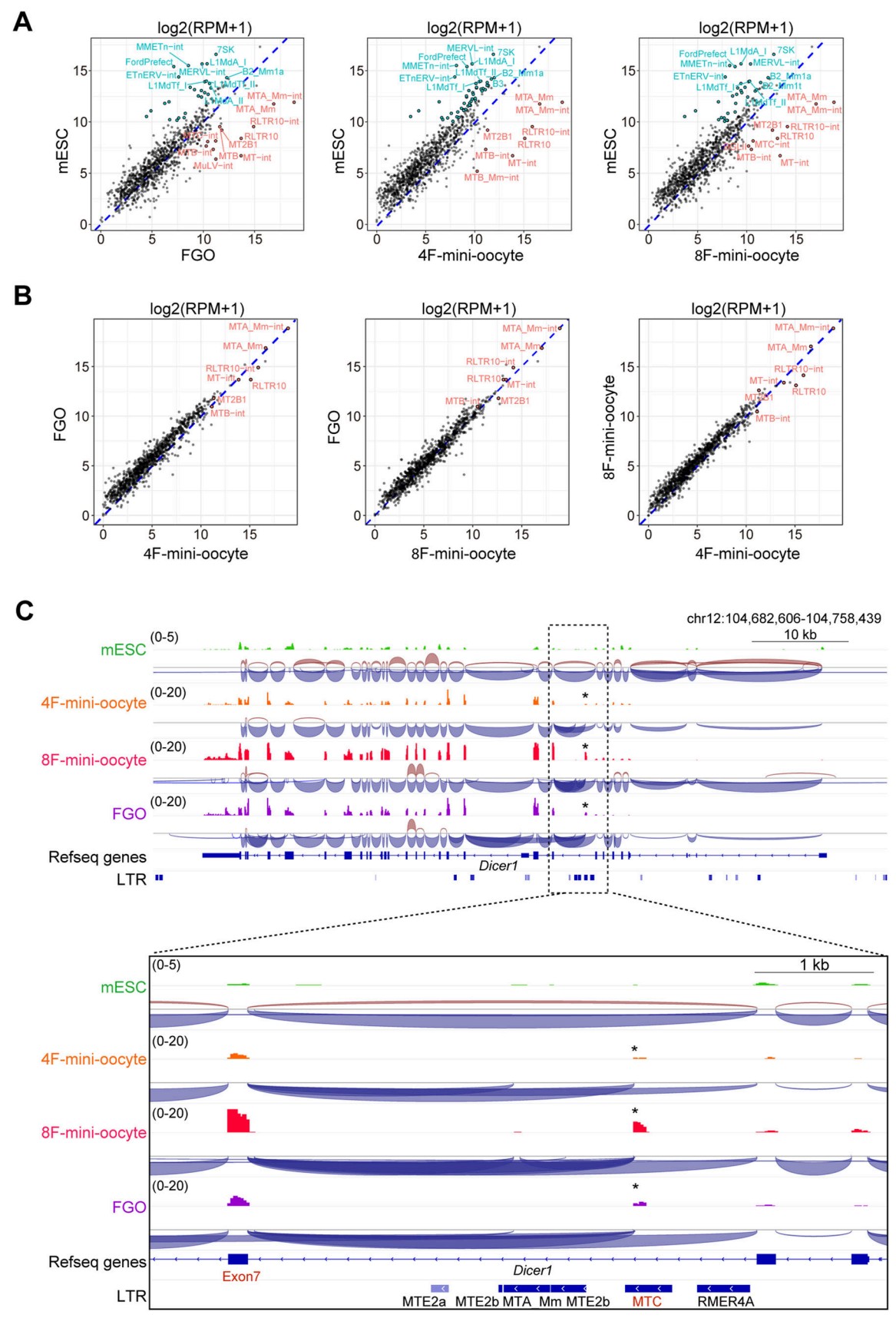

excessive DNA methylation, thereby regulating the developmental potential after fertilization (Li et al, 2018). Interestingly, although UHRF1 was localized in the nucleus of mESCs, it became exclusively localized in the cytoplasm in mini-oocytes upon induction (Fig 6B). To examine whether this change in localization is indeed dependent on DPPA3, we depleted DPPA3 using the CRISPR-Cas9 system (Fig S5A). Mini-oocytes could be generated normally upon DPPA3 depletion. As expected, the cytoplasmic localization of UHRF1 was lost, and UHRF1 was localized to the nucleus, as observed before in DPPA3 mutant mice (Li et al, 2018) (Fig 6B and C). These results indicate that DPPA3-dependent regulation of UHRF1 localization is operative in mini-oocytes as in in vivo oocytes.

We next explored the utility of mini-oocytes for genetic screening, focusing on the SCMC, which is specifically expressed in oocytes. The SCMC was first described as a complex composed of at least four members: Nlrp5 (also known as Mater), Tle6, Ooep (also known as Floped), and Khdc3 (also known as Filia). In addition to these, Padi6 has been proposed as a fifth member of the SCMC (Li et al, 2008). Female mice lacking SCMC genes do not exhibit defects in oogenesis but display embryonic arrest after fertilization (Tong et al, 2000; Esposito et al, 2007; Li et al, 2008; Yurttas et al, 2008; Zheng & Dean, 2009; Tashiro et al, 2010). Consistent with observations in mice, mutations in SCMC genes are associated with embryonic developmental arrest and female infertility in humans (Alazami et al, 2015; Zhu et al, 2015; Xu et al, 2016; Yatsenko & Rajkovic, 2019). Furthermore, recent evidence indicates that several mutations in SCMC genes cause multi-locus imprinting disturbances (MLID), suggesting their involvement in the regulation of DNA methylation (Docherty et al, 2015; Soellner et al, 2017; Elbracht et al, 2020; Bebbere et al, 2021). However, the global understanding of how SCMC genes influence DNA methylation in oocytes remains unclear, including whether all SCMC components are equally important or whether specific factors play unique roles. Leveraging the relative ease of gene manipulation in the mini-oocyte induction system, we examined whether the oocyte-specific cytoplasmic localization of UHRF1 and DNMT1 is affected by conducting a small-scale gene knockout screen for each SCMC member (Fig S5B). In line with the observation that female mice deficient in SCMC genes show normal oogenesis, there were no obvious alterations in generating mini-oocytes. This analysis revealed that only the knockout of Padi6, and not that of the other four members, affects the localization of UHRF1 (Fig 6D). In Padi6 knockout mini-oocytes, nuclear localization of UHRF1 became evident, although UHRF1 was still detectable in the cytoplasm (Figs 6D and E and S5C). Furthermore, both the nuclear and cytoplasmic UHRF1 signals were reduced upon Padi6 depletion (Fig S5D), indicating the down-regulation of UHRF1 protein levels. This was confirmed by

Western blotting for UHRF1 (Figs 6F and S5E). On the other hand, DNMT1 localization was not affected (Figs 6G and H and S5D). These findings align with a recent study that identified UHRF1 relocalization and reduction in Padi6 mutant mouse oocytes (Giaccari et al, 2024). On the other hand, whether the absence of other SCMC members affects UHRF1 localization in in vivo oocytes remains to be investigated. Therefore, our results suggest that although SCMC members are interdependent in the formation of the complex, the depletion of individual members does not lead to a uniform phenotype. Instead, each member or Padi6 may have distinct and unique functions within oocytes. This is consistent with the observation that Padi6 and Tle6 mutant mice exhibit differential phenotypes in the alterations of proteomic profiles (Jentoft et al, 2023).

## Discussion

In this study, we developed a method to generate mini-oocytes from mESCs. This method uses a conventional two-dimensional culture and only requires a medium change every 3 d during mini-oocyte induction. In addition, the scale of culture can be easily adjusted based on the desired number of mini-oocytes. Thus, this approach overcomes the limitations inherent in in vivo studies. Importantly, we observed that several key processes characteristic of in vivo oocyte development also occur during mini-oocyte induction. One of the prominent features of the oocyte is its large cell size, which is required for embryogenesis. During the mini-oocyte induction, cells exhibited an increase in size, resembling the growth phase of oocytes following the activation of primordial follicles. Our results indicated that supplementation of SCF in the culture medium is essential for this increase in cell size. It has been proposed that SCF binds to KIT, its receptor, and activates the PI3K (phosphoinositide 3-kinase)-AKT signaling pathway (Castrillon et al, 2003; Liu et al, 2006). AKT phosphorylates FOXO3, which inhibits its nuclear retention and promotes its cytoplasmic localization. However, the mechanism by which FOXO3 relocalization leads to oocyte activation and subsequent cell growth remains to be understood. Our findings suggest that the SCF-dependent pathway may regulate cell size during oocyte development. When combined with biochemical assays and genetic manipulations, which are relatively easily applicable to mini-oocytes, the mini-oocyte induction system might be able to facilitate the identification of regulatory molecules within the PI3K-AKT-FOXO3 pathway, as well as elucidate the molecular mechanisms underlying cell size increase during oogenesis.

We investigated whether mini-oocytes could serve as a model to conduct a genetic screen, using UHRF1 localization as a readout.

**Figure 3. Activation of oocyte-specific transposable elements in mini-oocytes.**
**(A)** Scatter plots showing the differential expression of transposable elements in in vivo fully grown oocytes (FGOs), 4F-mini-oocytes, and 8F-mini-oocytes compared with mouse embryonic stem cells. Expression levels are shown by $\log_2$(RPM + 1). Transposable elements differentially expressed are highlighted. **(B)** Scatter plots showing the expression of transposable elements in in vivo FGOs, 4F-mini-oocytes, and 8F-mini-oocytes. Expression levels are shown by $\log_2$(RPM + 1). Differentially expressed transposable elements are highlighted (see the Materials and Methods section for the cutoff used). **(C)** Genome browser snapshots showing RNA-seq data from mouse embryonic stem cells, 4F-mini-oocytes, 8F-mini-oocytes, and FGOs. Splicing patterns are visualized by Sashimi plots. Asterisks indicate reads mapped to the MTC element in the intron 6.

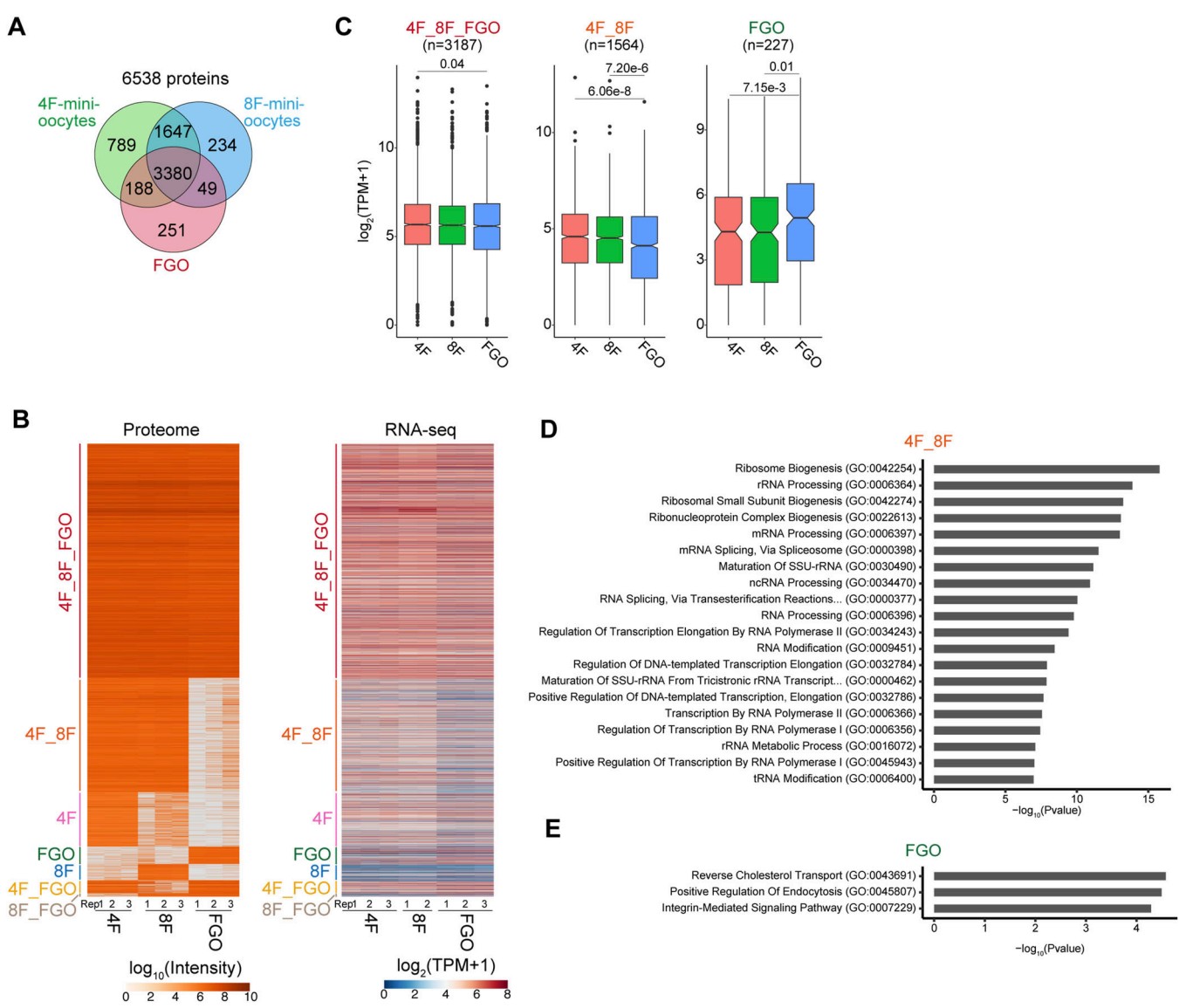

**Figure 4. Mini-oocytes exhibit proteomic profiles similar to in vivo oocytes.**
**(A)** Venn diagrams showing the overlap of expressed genes. Genes detected by proteome analysis of in vivo fully grown oocytes (FGOs), day 21 4F-mini-oocytes, and day 21 8F-mini-oocytes were compared. **(B)** Heatmaps showing the protein (left) and transcript levels (right) of different gene groups. 4F_8F_FGO: genes commonly expressed in 4F- and 8F-mini-oocytes and FGOs, based on proteomic data. 4F_8F: genes expressed in 4F- and 8F-mini-oocytes but not in FGOs. 4F: genes expressed exclusively in 4F-mini-oocytes. 8F: genes expressed exclusively in 8F-mini-oocytes. FGO: genes expressed exclusively in FGOs. 4F_FGO: genes expressed in 4F-mini-oocytes and FGOs, but not in 8F-mini-oocytes. 8F_FGO: genes expressed in 8F-mini-oocytes and FGOs, but not in 4F-mini-oocytes. **(C)** Boxplots showing the transcript levels of different gene groups. **(B)** Gene groups indicated correspond to those indicated in (B). The Kruskal–Wallis test and Dunn's test were applied. **(D)** Gene ontology analysis. Genes expressed in 4F- and 8F-mini-oocytes, but not in FGOs (4F_8F), were analyzed. **(E)** Gene ontology analysis. Genes expressed in FGOs, but not in 4F- and 8F-mini-oocytes (FGO), were analyzed.

During mini-oocyte induction, UHRF1 was relocalized to the cytoplasm, mimicking its behavior in in vivo oocytes (Maenohara et al, 2017). When DPPA3 was knocked out, UHRF1 exhibited nuclear localization in mini-oocytes, consistent with observations in DPPA3 knockout mice (Li et al, 2018). These findings strongly suggested that the molecular network intrinsic to in vivo oocytes is at least partially reconstituted in mini-oocytes. We then examined whether the role of SCMC components in regulating UHRF1 localization could also be studied using mini-oocytes. As a result, we found that PADI6, but not other SCMC components,

affects UHRF1 localization. Interestingly, Padi6 and Tle6 mutant oocytes showed distinct changes in the proteome, and UHRF1 was down-regulated specifically in Padi6 mutant oocytes, whereas both mutants showed impaired cytoplasmic lattice formation (Esposito et al, 2007; Qin et al, 2019; Jentoft et al, 2023). Furthermore, nuclear UHRF1 localization in Padi6 mutant mouse oocytes was consistently observed in a previous study (Giaccari et al, 2024). Thus, our results strongly suggest that mini-oocytes could serve as a useful tool for screening and selecting candidate genes before undertaking time- and cost-intensive in vivo studies. In addition,

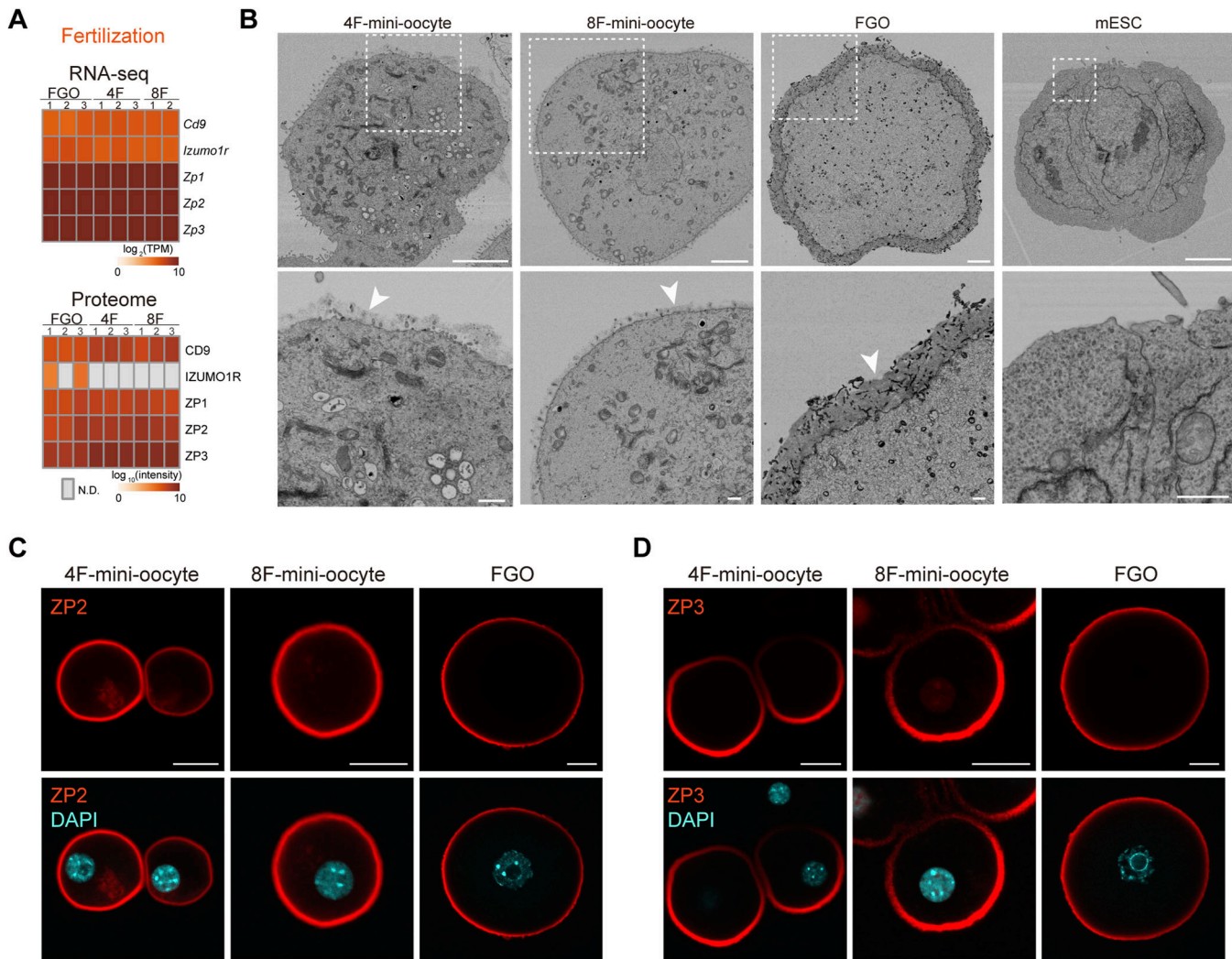

**Figure 5. Zona pellucida–like structures are formed in mini-oocytes.**
**(A)** Heatmaps showing the expression of fertilization-related genes based on RNA-seq or proteome analysis. Gray squares in the heatmap indicate that their expression was not detected (N.D.). **(B)** Electron micrographs of 4F- and 8F-mini-oocytes, fully grown oocytes, and mouse embryonic stem cells. The regions indicated by dashed squares are magnified at the bottom. Scale bars, 5 $\mu m$ (top) and 1 $\mu m$ (bottom). **(C)** Immunofluorescence for ZP2. DAPI was used to stain DNA. Scale bars, 20 $\mu m$. **(D)** Immunofluorescence for ZP3. DAPI was used to stain DNA. Scale bars, 20 $\mu m$.

the fact that more than $10^6$ mini-oocytes can be obtained from a single experiment suggests that genome-wide genetic screening and many kinds of biochemical assays are feasible using mini-oocytes.

The most prominent difference between the induction of mini-oocytes and in vivo oogenesis is the absence or presence of ovarian somatic cells. In vivo, oocytes are surrounded by ovarian somatic cells, particularly granulosa cells. It is well established that cell–cell communication between oocytes and somatic cells, such as cell–cell contact by transzonal projections and SCF secretion from granulosa cells, plays a crucial role in oocyte development (Liu et al, 2006; Li & Albertini, 2013). In developing the mini-oocyte induction method, we chose not to include ovarian somatic cells, prioritizing the method's simplicity and scalability. As a result, it is highly likely that certain molecular processes dependent on oocyte–somatic cell communication are absent in

the mini-oocyte induction system, which may limit the utility of mini-oocytes as an oocyte model. It is apparent that mini-oocytes are smaller than in vivo FGOs. Our finding that genes involved in the cAMP signaling pathway were down-regulated in mini-oocytes may be informative for the improvement of their size, as it is related to follicle formation and meiotic arrest (Wang et al, 2015). In addition, the zona pellucida was poorly developed and cytoplasmic lattices were not detectable in mini-oocytes, although we cannot exclude the possibility that a small fraction of mini-oocytes might develop these structures. These structural differences might be involved in the presence of somatic cell–derived factors, although this hypothesis needs to be addressed by further experiments. Of note, a study that generated oocyte-like cells in vitro from mESCs without ovarian somatic cells has been reported very recently (Nosaka et al, 2025). Although this study allowed the progression of meiosis followed by oocyte-like cell generation

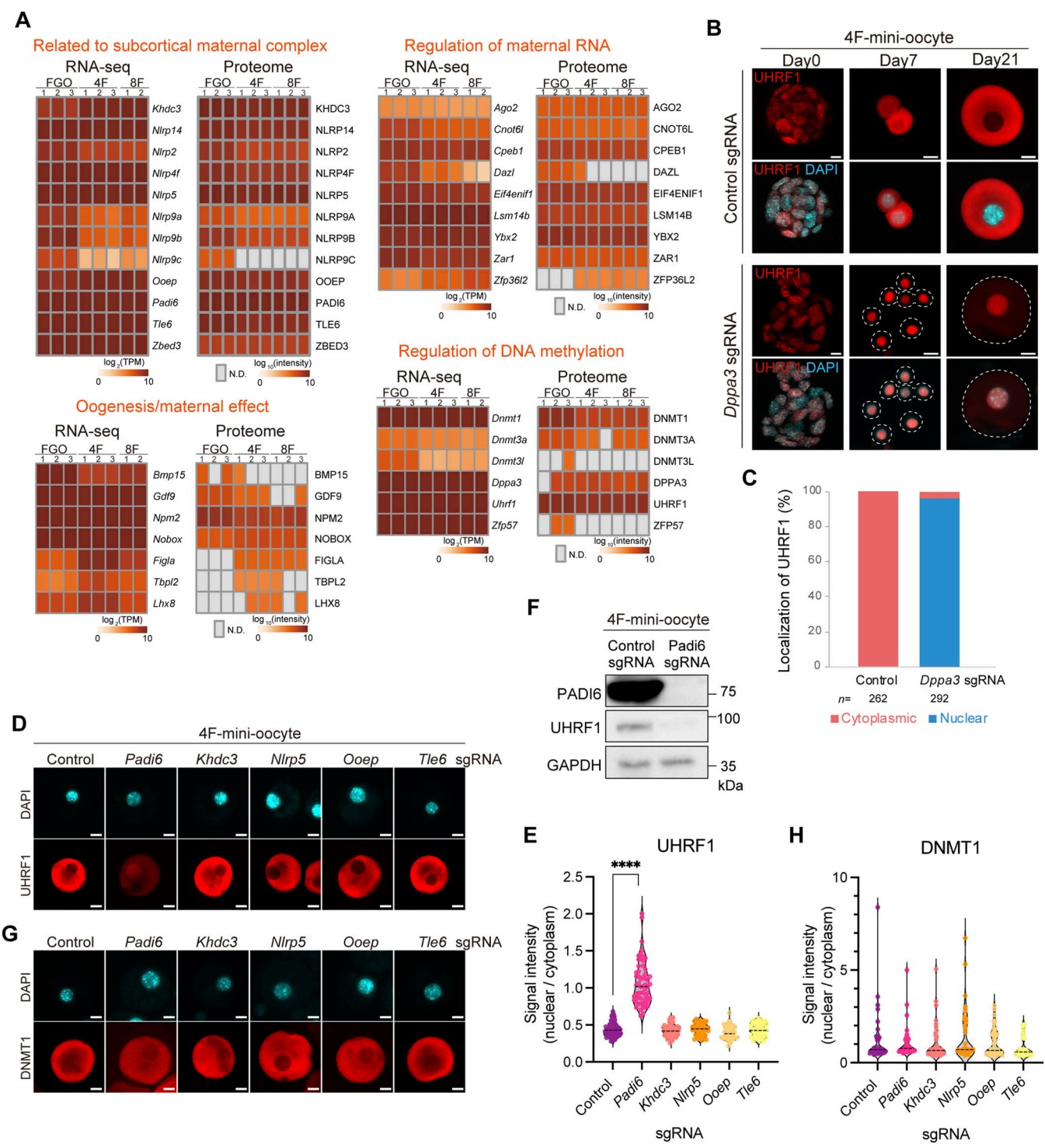

**Figure 6. Small-scale gene knockout screen using mini-oocytes.**
**(A)** Heatmaps showing the expression of indicated genes based on RNA-seq or proteome analysis. Genes characteristic of oocytes are grouped according to their known functions. Note that the expression of Figla, Tbpl2, Nobox, and Lhx8 includes both endogenous and transgene expressions. Gray squares in the heatmap indicate that their expression was not detected (N.D.). **(B)** Immunofluorescence for UHRF1 in 4F-mini-oocytes. Immunofluorescence was performed for cells transduced with control sgRNA or Dppa3 sgRNA at days 0, 7, and 21 after the mini-oocyte induction. Scale bars, 10 $\mu$m. **(C)** Bar graph showing the frequency of cells displaying nuclear (blue) and cytoplasmic (red) localization of UHRF1 in cells transduced with control sgRNA or Dppa3 sgRNA. $n$, total cell numbers analyzed in three independent experiments. **(D)** Immunofluorescence for UHRF1 in 4F-mini-oocytes at day 21. Immunofluorescence was performed for cells transduced with indicated sgRNA. Scale bars, 10 $\mu$m. **(E)** Violin plots showing the nuclear and cytoplasmic UHRF1 signal intensity ratio in day 21 4F-mini-oocytes transduced with indicated sgRNA. **(F)** Western blotting for PADI6, UHRF1, and GAPDH in 4F-mini-oocytes at day 21. Indicated sgRNAs were transduced. **(G)** Immunofluorescence for DNMT1 in 4F-mini-oocytes at day 21. Immunofluorescence was performed for cells transduced with indicated sgRNA. Scale bars, 10 $\mu$m. **(H)** Violin plots showing the nuclear and cytoplasmic DNMT1 signal intensity ratio in day 21 4F-mini-oocytes transduced with indicated sgRNA.

upon 34 d of culture of sorted PGCLCs, the size of the oocyte-like cells was still smaller than in vivo FGOs. In this regard, incorporating somatic cells or conditioned medium into the mini-oocyte induction system may help distinguish between cell-autonomous and non–cell-autonomous pathways in constructing oocyte-specific features.

# Materials and Methods

### Animals and collection of mouse oocytes

All animal experiments were approved by the Animal Experiments Committee of the University of Yamanashi (A4-1), and performed according to the guidelines for animal experiments at the University of Yamanashi. Mice were housed in cages under specific pathogen-free conditions and had free access to water and food. FGOs were collected by poking the ovaries of ICR female mice at 8–12 wk of age. For proteome analysis, 5 IU of pregnant mature serum gonadotropin (PMSG) was injected into B6D2F1 female mice 46–48 h before the collection of FGOs.

### Cell culture

Female mESCs were cultured in the 2i+LIF condition without feeders as described previously (Hayashi et al, 2011). BVSC H18 mESCs (Hayashi et al, 2012) were used as parental cells for 4F- and 8F-mESCs.

### Mini-oocyte induction

0.01% poly-L-ornithine solution (Sigma-Aldrich) was added to each well of a plate and incubated at RT for 1 h. Subsequently, poly-L-ornithine solution was removed, 300 ng/ml laminin solution (Corning) was added to the same wells, and the plate was incubated overnight in a 5% $CO_2$ incubator. The wells were washed once with PBS, and S10 medium (StemPro-34 SFM [Thermo Fisher Scientific] supplemented with 10% FBS [Thermo Fisher Scientific], 150 $\mu$M 2-O-$\alpha$-D-glucopyranosyl-L-ascorbic acid [TCI chemicals], 1 × GlutaMAX [Thermo Fisher Scientific], 1 × penicillin/streptomycin [Thermo Fisher Scientific], and 55 $\mu$M 2-mercaptoethanol [Thermo Fisher Scientific]) supplemented with 150 ng/ml SCF (R&D), 10 $\mu$M Y-27632 (FujiFilm), and 0.5 $\mu$M Shield-1 (Selleck) was added to the plate to prepare it for cell seeding.

4F- or 8F-mESCs routinely cultured were detached from culture plates by 1× TrypLE solution (Thermo Fisher Scientific), and the detached cells were thoroughly dissociated into single cells by pipetting. The cell suspension was then transferred to tubes containing Wash medium (DMEM/F12 [Nacalai] supplemented with 0.1% BSA [Thermo Fisher Scientific]). The cell suspension was centrifuged at 200$g$ for 3 min at RT, and the supernatant was discarded. The cell pellet was resuspended in an appropriate volume of S10 medium supplemented with SCF, Y-27632, and Shield-1. The cell suspension was added to the precoated culture plates in the desired amount, ensuring even distribution of cells (regarded as day 0). $3 \times 10^4$ cells were typically plated per well in a 24-well plate. The medium was replaced every 72 h, with half of the medium exchanged for freshly prepared S10 medium containing SCF, Y-27632, and Shield-1. Starting from day 12, Shield-1 was omitted from the fresh medium.

### Gene knockout in mini-oocytes

A 4F-mESC clone stably expressing Cas9 (Cas9-mESCs) was established using a lentivirus vector carrying the CAG-Cas9-2A-Blast cassette (a plasmid kindly provided by Yusuke Miyanari, Kanazawa University, Japan). In brief, 4F-mESCs were infected with Cas9 lentivirus and selected with blasticidin, and a single blasticidin-resistant clone that efficiently expresses Cas9 was used for transfection. For sgRNA expression, oligos containing sgRNA target sequences were first inserted into a constructed sgRNA-expressing vector that confers zeocin resistance (U6-sgRNA-CAG-Zeo). At day 0, the sgRNA-expressing vector was transfected into Cas9-mESCs using Lipofectamine 2000 (Thermo Fisher Scientific), and the medium was exchanged with a fresh one 5–6 h after the transfection. From day 1, the transfected cells were selected with 50 $\mu$g/ml zeocin for 48 h. After the selection, mESCs were cultured for 24–48 h without zeocin, and then used for mini-oocyte induction. We used a pair of sgRNAs to efficiently knock out a target gene. The sgRNA target sequences are as follows:

Negative control #1: AAACCTAGCGTAGATTCGGC.
Negative control #2: CAATATCTAAGCGCTAACGA.
Dppa3 #1: TGCCCATCGCATCGCCATGG.
Dppa3 #2: TGCGGTTCCGTAGACTGCGC.
Padi6 #1: GGTAGGCATGGAAATCACCT.
Padi6 #2: GAGGCAGACATCTATCGAGA.
Nlrp5 #1: GTTACCGGTTTGCAGAAAGT.
Nlrp5 #2: CTGAAGGAGTGTGGACCATG.
Tle6 #1: ACCTCAGAGGTGGTGCGTGA.
Tle6 #2: CCAAGTTCAAAAGCACCCCG.
Ooep #1: GGCGTCAGCATCAGCCGTGT.
Ooep #2: CAGTGGACTCCGTCAACTCT.
Khdc3 #1: ACACTATGGCCTCTCTGAAG.
Khdc3 #2: TCAACGTCCCTTCGGAAAGG.

### Quantitative real-time polymerase chain reaction (qRT–PCR)

qRT–PCR for mini-oocytes at day 21 was performed as described before (Sakamoto et al, 2022). To estimate knockout efficiency, primers that overlap regions possibly deleted by paired sgRNA were designed. The primers used were as follows:

Padi6-F: 5'-actgtggatgaagacaaggtgc
Padi6-R: 5'-atgtccagttgtccatctcg
Nlrp5-F: 5'-agactctcaccggtttgtatg
Nlrp5-R: 5'-catagaacaccgacctcatgg
Khdc3-F: 5'-acgtggaacctcggctactg
Khdc3-R: 5'-gtcaacgtcccttcggaaag
Tle6-F: 5'-ggagaggaacaagatgagtattg
Tle6-R: 5'-cttttgaacttggggtgctg
Ooep-F: 5'-tgctgacgccaagccagact
Ooep-R: 5'-ttgacggagtccactgtcag
Stella-F: 5'-aagaggacgctttggatgat
Stella-R: 5'-aatgcggttccgtagactg

## Transcriptome analysis

1,000 and 10 mini-oocytes at days 7 and 21, respectively, were pooled in PBS per sample and snap-frozen in PCR tubes. SMART-seq Stranded Kit (TAKARA) was used to prepare RNA-seq libraries. The first and second amplification steps were carried out for 5 and 13 cycles, respectively. Paired-end sequencing was performed on a NovaSeq 6000 (53 bp × 2) or NovaSeq X Plus (150 bp × 2).

## RNA-seq data processing

For samples prepared using a SMART-seq stranded kit (TAKARA), the first three bases of read 2 derived from the SMART-seq stranded adaptor were removed using Trim Galore! with a "–clip_R2 3" option. After the reads were aligned to the mouse genome (mm10) using Hisat2, the reads that correspond to rRNA were removed using the Bedtools intersectBed function (Quinlan & Hall, 2010). The uniquely mapped reads were isolated using the tag NH:i:1. featureCounts (Liao et al, 2014) and GENCODE annotation of the mm10 mouse genome (vM25) were used to analyze the transcription level with options "-p -O --fraction." Published total RNA-seq data for FGOs (GSE183969) or mRNA-seq data for differentiating oocytes (GSE79729 and GSE128305) were used to analyze their transcriptional profiles (Hikabe et al, 2016; Shimamoto et al, 2019; Yano et al, 2022). TPM was calculated by counting reads mapped to the exon. PCA was performed with R. The resulting read count data were processed using EdgeR (Robinson et al, 2010) to identify differentially expressed genes. A false discovery rate of <0.05 was used to extract differentially expressed genes. KEGG pathway enrichment analysis was conducted using the ClueGO plugin in Cytoscape (Shannon et al, 2003). For quantification of repetitive genomic regions, we used featureCounts with options "-M -p -s 0" to allow for multi-mapped read counts of repetitive loci annotated by RepeatMasker (mm10). The expression pattern of transposable elements in mESCs was calculated using total RNA-seq data from male mESCs (the identifiers were ENCLB212CDP and ENCLB118KZE) (https://www.encodeproject.org) as corresponding data for female mESCs were not available in the database. Repetitive elements with more than 100 copies were used, and reads per million (RPM) were calculated using R. The mean of replicates was calculated, and repetitive elements with $\log_2(RPM + 1) > 10$ and absolute $\log_2$ fold change ≥ 2 were defined as differentially expressed.

## Proteome analysis

### Sample preparation

100 in vivo FGOs and ~$10^4$ mini-oocytes per sample were collected in PBS containing 0.2% PVP. Zona pellucida of FGOs was removed by acidic Tyrode's solution. The cell pellets were lysed in PTS buffer (12 mM sodium deoxycholate [SDC], 12 mM sodium N-lauroylsarcosinate [SLS] in 100 mM Tris–HCl buffer [pH 8.5]) with 0.02% lauryl maltose neopentyl glycol (LMNG) (Konno et al, 2024). Proteins were reduced and alkylated using 12 mM tris(2-carboxyethyl)phosphine (TCEP) and 50 mM 2-chloroacetamide (CAA). The protein solution was diluted 10-fold with 50 mM ammonium bicarbonate and digested with 0.5 mg LysC (FUJIFILM Wako, biochemistry-grade) and 0.5 mg trypsin (Promega, sequence-grade) at RT overnight. Digestion was halted by acidifying the mixture with trifluoroacetic acid (TFA). The resulting peptides were purified using SDB-RPS StageTips (GL Sciences).

### Nanoscale liquid chromatography/tandem mass spectrometry (nanoLC/MS/MS)

A nanoLC/MS/MS system comprising a Vanquish Neo UHPLC (Thermo Fisher Scientific) and an Orbitrap Eclipse mass spectrometer (Thermo Fisher Scientific) equipped with a high-field asymmetric waveform ion mobility spectrometer interface (FAIMSpro, Thermo Fisher Scientific) was employed. The mobile phases consisted of (A) 0.1% formic acid and (B) 0.1% formic acid and 80% acetonitrile (ACN). In-house–packed columns were prepared as follows: emitters were generated by pulling a 25-cm fused-silica capillary (100 $\mu$m inner diameter; GL Sciences) using the P-2000 laser puller (Sutter Instrument). ReproSil-Pur C18-AQ resin (1.9 $\mu$m, Dr. Maisch) was then packed into the emitter using an air-pressure pump connected to a $N_2$ bomb, generating a 15-cm column. Peptides were separated by applying a linear gradient for 45 min (5–40% B over 45 min, 45–99% B over 5 min, and 99% B for 10 min) at the flow rate of 350 nl/min. All spectra were obtained using the Orbitrap analyzer. Data-independent acquisition (DIA) mode was employed with the following parameters. Survey scans were performed in the range of 350–1,000 $m/z$ (resolution = 120,000, maximum injection time = 45 ms, and automatic gain control = 300%). In the following MS/MS scans, the precursor range was set to 500–740 $m/z$, and 60 scans were acquired with the isolation window of 4 $m/z$, with HCD normalized collision energy of 27 (resolution = 15,000, injection time = 22 ms, auto gain control = 1,000%, first mass = 120 $m/z$). FAIMS CV was fixed to –45.

### Data processing

The raw data files were processed using DIA-NN v1.9.1 (Demichev et al, 2020). A spectral library for the precursor ion identifications was made with DIA-NN using an UniProt Swiss-Prot/TrEMBL FASTA file database (*Mus musculus*) downloaded on February 2024, and predicted FASTA digest with default settings. Then, the raw data files were searched using the spectral library with default parameters at 1% FDR. Intensity values were normalized using the MaxLFQ algorithm (Cox et al, 2014) implemented in DIA-NN based on the fact that most of the proteome typically does not change between any two conditions so that the average behavior can be used as a relative standard. However, it is important to note that the sensitivity of protein detection may differ between mini-oocyte and FGO samples because of differences in input amounts.

Proteins consistently detected in all three replicates were used to analyze the overlap of proteins among the FGO, 4F-, and 8F-mini-oocyte groups by depicting a Venn diagram. Heatmaps were generated using the pheatmap function in the R package based on normalized intensity data. Gene ontology analysis was performed using enricher (Chen et al, 2013; Kuleshov et al, 2016; Xie et al, 2021).

## Electron microscopy

Cells were fixed in 2% PFA and 2.5% glutaraldehyde in 0.1 M sodium cacodylate buffer, pH 7.4, for 2 h at RT and stored at 4°C. After

washing cells with 0.1 M cacodylate buffer (pH 7.4) three times, they were further fixed with 1.5% potassium ferrocyanide and 1% osmium tetroxide in the same buffer at 4°C for 1 h and then washed five times with Milli-Q water. After the samples were immersed in 0.5% uranyl acetate in Milli-Q water, they were dehydrated with an ethanol series (30%, 70%, 85%, 95%, and 99.5% for 15 min each and three times in 100% for 20 min) and propylene oxide for 20 min twice, followed by infiltration with Epon 812 resin (TAAB). Polymerization was performed at 65°C for 72 h. Sections (200 nm) were cut with a diamond knife on an ultramicrotome (Leica EM UC6, Leica Microsystems) and placed on a piece of silicon wafer. The sections were stained with uranyl acetate and lead nitrate, then observed with a field emission scanning electron microscope JSM-IT800SHL (JEOL) at 3 kV using a back-scattered electron detector.

### Immunofluorescence and Western blotting

FGOs and mini-oocytes were immunostained as described previously (Ishiuchi et al, 2021). In brief, cells were fixed with 4% PFA in PBS for 10 min at RT. The fixed cells were permeabilized with 0.2% Triton X-100 in PBS for 15 min at RT and incubated in a blocking buffer (3% BSA in PBS) for 1 h. The cells were then incubated with primary antibodies overnight at 4°C or for 1 h at RT. The antibodies used are anti-UHRF1 (sc-373750, 1:200; Santa Cruz Biotechnology), anti-MVH (ab27591, 1:200; Abcam), anti-FOXO3 (#2497, 1:400; CST), anti-DNMT1 (ab188453, 1:200; Abcam), anti-ZP2 (IE-3, 1:200), and anti-ZP3 (IE-10, 1:200) (East & Dean, 1984; East et al, 1985). Anti-ZP2 and ZP3 antibodies were kindly provided by Masahito Ikawa (Osaka University). Secondary antibodies used are goat Alexa 555–conjugated anti-rat IgG or donkey Alexa 555–conjugated anti-mouse IgG (1:500; Thermo Fisher Scientific) or anti-rabbit IgG (Abcam). The samples were mounted with VECTASHIELD (Vector Laboratories). Images were acquired by FV1200 confocal microscope (Olympus, objective: UPlanSApo 40x/0.95) or Axioscope 5 epifluorescence microscope (ZEISS, objective: N-Achroplan 40x/0.65). To compare signal intensities between samples in a single experiment, parameters for the image acquisition were first set avoiding the saturation of the signals, and the same setting was kept to acquire images for all the samples. ZP2- and mCherry-positive and mCherry-negative cells were manually counted by evaluating fluorescence intensity relative to the background signal. Whole-cell lysates of induced mini-oocytes at day 21 were used for Western blotting. Anti-UHRF1 (sc-373750, 1:1,000; Santa Cruz Biotechnology), anti-PADI6 (LS-C373394, 1:1,000; LS Bio), and anti-GAPDH (60004-1-Ig, 1:1,000; Proteintech) antibodies were used.

### Fluorescence intensity measurement

Nuclear regions were first determined by thresholding of DAPI staining images using Fiji. After the average fluorescence intensity at the nuclear regions was measured, the nuclear regions were discarded from the image, and the signal intensity at the cytoplasmic regions was measured. Background signal intensities were subtracted during data analysis. The data obtained from

independent experiments were combined after the examination of their reproducibility.

### Statistical analysis

Statistical analyses were implemented with GraphPad Prism or R software (http://www.r-project.org).

## Data and Code Availability

The sequencing data from this study are available at the Gene Expression Omnibus, accession number GSE290666. The proteomic data have been deposited into the ProteomeXchange Consortium via the jPOST (Okuda et al, 2025) partner repository, with the dataset identifier JPST003661 (PXD061359 for ProteomeXchange). Original data are available upon reasonable request. Please contact tishiuchi@yamanashi.ac.jp.

## Supplementary Information

## Acknowledgements

We thank the members of our laboratory at the University of Yamanashi for discussion and Masahito Ikawa (Osaka University) for technical support. This work was supported by grants from MEXT Grant-in-Aid for Scientific Research on Innovative Areas (JP19H05756 to T Ishiuchi) and for Transformative Research Areas (JP25H01354 to T Ishiuchi), Takeda Science Foundation (to T Ishiuchi), Mochida Memorial Foundation for Medical and Pharmaceutical Research (to T Ishiuchi), Naito Foundation (to T Ishiuchi), Uehara Memorial Foundation (to T Ishiuchi), the Japan Science and Technology Agency (JPMJMS2022 to S Yonemura), and the Japan Society for the Promotion of Science (24K02228 to S Yonemura; 22K15125 and 24K01938 to K Shirane; and 23H04949, 24H00059, and 23K20043 to K Hayashi). This work was also supported by the COI-NEXT Support Unit for Imaging Science at Kento and World Premier International Research Center Initiative (WPI), MEXT, Japan.

### Author Contributions

A Banno: formal analysis, investigation, and methodology.
K Mizuno: formal analysis, investigation, and methodology.
M Sakamoto: formal analysis.
S Komatsubara: investigation.
K Shirane: resources and writing—review and editing.
K Hayashi: resources and writing—review and editing.
N Hamazaki: resources.
K Imami: formal analysis, investigation, and writing—review and editing.
S Yonemura: investigation and writing—review and editing.
T Ishiuchi: conceptualization, supervision, funding acquisition, project administration, and writing—original draft, review, and editing.

### Conflict of Interest Statement

N Hamazaki is a co-founder of Dioseve, and the other authors have no conflicts of interest.

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
