## [Reviewer comments · Life Science Alliance]

A simple, efficient, and scalable method to generate oocyte-like cells in vitro

Ami Banno, Kana Mizuno, Mizuki Sakamoto, Sota Komatsubara, Kenjiro Shirane, Katsuhiko Hayashi, Nobuhiko Hamazaki, Koshi Imami, Shigenobu Yonemura, and Takashi Ishiuchi

DOI: <https://doi.org/10.26508/lsa.202503379>

Corresponding author(s): Takashi Ishiuchi, University of Yamanashi

Review Timeline:

Submission Date:	2025-05-02
Editorial Decision:	2025-07-08
Revision Received:	2025-09-25
Editorial Decision:	2025-11-04
Revision Received:	2025-11-07
Accepted:	2025-11-07

Scientific Editor: Sarita Hebbar

Transaction Report:

July 8, 2025

Re: Life Science Alliance manuscript #LSA-2025-03379-T

Dr. Takashi Ishiuchi
University of Yamanashi
Faculty of Life and Environmental Sciences
4-4-37 Takeda
Kofu, Yamanashi 400-8510
Japan

Dear Dr. Ishiuchi,

Thank you for submitting your manuscript entitled "Establishment of a simple, efficient, and scalable method to generate oocyte-like cells in vitro" to Life Science Alliance. The manuscript was assessed by two expert reviewers, whose comments are appended to this letter.

Overall both reviewers found this extensive characterisation of mini-oocytes interesting and of potential value to the field. They have also raised important concerns that need to be addressed before publication at LSA.

Reviewer 1 expressed concern that the work is missing information on purity of cultures in relation to percentage of oocytes. In fact both reviewers have carefully specified aspects in the methods and results sections that could be more accurately described. We concur that all these requested clarification points, including for purity of cultures, must be addressed in the revised submission.

Reviewer 2 has suggested inclusion of a different normalisation step for the proteomics data, that we concur with.

Both reviewers also indicated that the manuscript must elaborate on the potential of these mini-oocytes (spermatozoa attaching to oocytes and potential for cleavage divisions). We agree that the revised version must attend to this, and to other comments that would help to better position this study in terms of its novelty and in relation to existing literature.

We invite you to submit a revised manuscript addressing the reviewers' comments. When submitting the revision, please include a letter addressing the reviewers' comments point by point. While a rebuttal must respond to all points in some form, additional experiments to resolve these points (other than ones indicated above) are not required.

Thank you for this interesting contribution to Life Science Alliance. We are looking forward to receiving your revised manuscript.

Sincerely,

Sarita Hebbar, PhD
Scientific Editor
Life Science Alliance
<http://www.lsjournal.org>

B. MANUSCRIPT ORGANIZATION AND FORMATTING:

Reviewer #1 (Comments to the Authors (Required)):

The manuscript entitled "Establishment of a simple, efficient, and scalable method to generate oocyte like cells in vitro" by Banno et al. advances on the methodology for differentiating mouse pluripotent stem cells into cells of the female germline. The authors adapt a previously reported the transcription factor (TF)-based oocyte-like cell reprogramming method. In contrast, to the original protocol by Hamazaki et al. (2020) a 2D culture system without the need for gonadal somatic cells is used for generating so called "mini-oocytes". It is impressive to see that the resulting oocytes develop without the formation of typical follicle structure suggesting an largely cell autonomous process. Considering that oocytes are available in very limited numbers from mice, the approach is notable as a large number of oocytes can be obtained in relatively pure cultures (albeit data on homogeneity is missing) that greatly would facilitate proteomic, molecular analysis, and genomic screening. The authors perform a extensive characterization including morphology, size, transcriptome and proteome analysis, and molecular features of their mini-oocytes. These data suggest that minioocytes might resemble some characteristics of oocytes of primordial follicles. However, similarity is limited and it is clear that important differences exist that might limit the usefulness of minioocytes. In particular, meiosis is not represented as well as there is no opportunity to study the functionality of the ooplasm as the stage where activation can be considered is not reached, and hence embryonic potential (at least cleavage divisions) cannot be assessed. Although, the advance is encouraging the present version appears preliminary as no new mechanistic insight is reached and the suggested application for studying processes of oogenesis remains somewhat speculative. The impact of the study could be considerably improved by demonstrating a mechanistic advance beyond what has already been observed in in vivo oocytes. Yet, the presented characterization appears to remain on a level showing resemblance of processes in minioocytes to known regulation from in vivo oocytes.

Specific points

1. The advance over the original publication of TF mediated reprogramming and other systems that do not use gonadal somatic cells could be better described. The 8 TFs have been reported and a brief explanation of the inducer SHIELD would aid the reader. A focus on innovation in the culture beyond omitting aggregation with gonadal somatic cells should be more clearly described. In particular, the culture relies on a number of previously suggested factors, and it is not clear what is new. Also it would be important to disclose why StemPRO has been used as for this culture medium it is unclear if the composition is known or commercially kept undisclosed. If the formulation would indeed be undisclosed this would lessen the enthusiasm considerably as biological understanding is likely limited. The authors should also consider the positioning of their minioocyte culture against earlier work of extended PGCLC culture that allows meiotic entry (Ohta et al. 2017, PMID: 28559416, Miyauchi et al., 2018).

2. The authors perform extensive characterization of their minioocytes and use these data to illustrate similarities to in vivo

oocytes. However, a number of important processes of oogenesis are not normal or represented in minioocytes including meiosis, formation of the zona pellucida, diameter remains below 40 micrometer. The lack of indication of functionality of the obtained minioocytes (activation, cleavage divisions) raise the concern that these might be poor approximations for many key processes and hence the question of applicability for future screening. A focus on presumably normal processes in minioocytes could help to understand future applications. A discussion of STRA8, ZGLP1 and FOXO3 expression data could also be useful for providing assessment. If an application for gaining novel mechanistic insight into one of these processes (Padi6 and STELLA regulation of UHRF1 are in the right direction but do not seem provide new insight against what is already known from in vivo oocytes) could be included, the study would more convincingly show that minioocytes are a useful model.

3. Meiosis is not initiated in minioocytes but these cells are still said to arrest. As this is considered an important process in oogenesis, it would be good to discuss mitotic arrest in minioocytes in relation to meiotic prophase arrest. Also, migratory PGCs undergo transcriptional arrest and (later also dormancy) - would these processes be observed similar in minioocytes?

4. One remarkable aspect of the culture system is that apparently a large fraction of surviving cells differentiate into minioocytes. However, this is stated in the text without any data about purity of the cultures cited. Could a more exact assessment be included what the percentage of oocytes and what other cell types (mesodermal?) in these cultures is?

Minor comments

- The protocol adjustments, e.g. media changes, Shield1 timing, are useful but do not constitute a fundamentally new method. The title should be revised to avoid overstating novelty and instead emphasize practical refinements for scalability or screening. The abstract and introduction should also be adjusted accordingly.
- P4, line 121 126 - Acknowledged that Shield1 as an inducer for overexpression of the PPT8 as described in Hamazaki et al., 2020, but maybe good to briefly explain the rationale.
- P5 - "Requirement of culture supplements for the generation of mini-oocytes" The culture conditions are also completely adapted from Hamazaki et al., 2020, except for the use of 10 μ M Y-27632 (vs. 10 nM in the original study). Was this concentration optimized here, or is it a discrepancy? Clarification is needed.
- P6 - The authors note that 4F mini-oocytes adopt a round morphology by day 7, suggesting a critical transition stage. The expression of early germline markers, e.g., MVH, DAZL, should be demonstrated at this stage. In fact, key timepoints (e.g., days 7 and 21) should be tracked to validate progression.
- P7 - The proteomic analysis and detection of ZP2/ZP3 (late-stage markers) are strengths. Are other key markers like NPM2 also expressed? And maybe point out in the main text as well if other key growth markers also expressing.
- Nobox is one of the inducible TFs the authors use for reprogramming. It should be clearly indicated that RNAseq and proteomics refers to the endogenous Nobox and not the transgenic factor.
- P25 - Figure 4, Proteome of mini-oocytes, at day 21 of culturing?
- P7, line 244 - Could the authors briefly speculate on why ribosomal biogenesis proteins are overexpressed in mini-oocytes, for example, is this a compensatory stress response, culture artifact, or arrested developmental state?
- P8, line 251- The title can be more precise, consider "Mini-Oocytes Partially Recapitulate the Ultrastructure of in vivo Oocytes"
- P8 - The authors note that cytoplasmic lattices are absent in mini-oocytes. Did proteomic analysis reveal whether related proteins are expressed or if required signaling pathways are missing?
- P8 - Validation of DDPA3 KO efficiency is missing
- P8 - Does DDPA3 KO only affect UHRF1 localization and not expression? unlike PADI6 loss?
- P9 - PADI6 loss affects UHRF1 expression and localization, maybe it would be useful to assess DNA methylation at a few loci e.g. imprinted genes as well? And similarly, for DDPA3.
- P9 - Is UHRF1 localization known to be affected by loss of other SCMC members in vivo?
- Since SCMC gene loss in mice causes multi-locus imprinting disturbances (MLID), it would be good to check whether other SCMC factors affect DNA methylation in mini-oocytes, even if UHRF1 localization is unchanged
- Do DDPA3 and PADI6 KOs alter mini-oocyte growth or morphology?
- The transcriptomic and proteomic data may reveal why mini-oocytes exhibit restricted size. Are there deficiencies in growth-promoting pathways? Highlighting these could guide future optimizations.
- In discussion, maybe good to suggest exploring additional growth signals to improve mini-oocyte size and maturation in future work for culture conditions.
- Overall, sub-headings and figure legends should be more detailed and concise.

Reviewer #2 (Comments to the Authors (Required)):

Given the technical complexities of differentiating pluripotent e.g. embryonic stem cells (ESC) into oocytes in vitro, the Authors aimed to provide a simplified protocol that allows to generate oocytes in approximately 3 weeks, via forced expression of transcription factors (TFs) in a two-dimensional culture method, in the mouse model. Specifically, the Authors used mouse ESC lines in which the expression of 4 TFs (Nobox, Figla, Tbp12, and Lhx8) or 8 TFs (the 4 TFs plus Stat3, Dynl1, Sub1, and Sohlh1) can be induced in the presence of Shield-1. After 3 weeks, the ESC-derived oocytes were small (approx. 40 microns, hence

called 'mini oocytes') but their transcriptomic and proteomic profiles were quite similar to those of oocytes fully grown in vivo. As proof of principle that the mini oocytes can be useful in studies of e.g. oocyte biology, CRISPR Cas9 technology was applied to inactivate genes of interest (Dppa3, Padi6, Nlrp5, Tle6, Ooep, Khdc3).

I have the following remarks.

How do the Authors explain that the zona pellucida was assembled outside the mini oocytes, although these were not encased in follicles? The ZP proteins are known to traffic independently of one another inside oocytes (PMID 17047254), and then are exocytosed. However, in the cell culture system used by the Authors, there is nothing that prevents the ZP proteins from diffusing away after they have been released outside. What was that kept the ZP proteins from diffusing away and allowed them instead to polymerize around the mini oocytes? Indeed, other secreted proteins like GDF9 and BMP15 diffuse away and are detected at low or minimal level in mini oocytes (Fig. 6A). Could the Authors please also add ZP1-2 to the heat maps of Fig.6A?

The heat maps in Fig.4B, Fig.5A and 6A could be easier to interpret.

Are the color-coded values referring to 100 FGOs and 10000 mini oocytes, or are they referring to single cells? How was the normalization performed?

In the transcriptome analysis the values shown in the heat maps are normalized (TPM, transcripts per million), but how have the results in the proteome analysis (data-independent acquisition mode) been normalized?

In my opinion the values should be presented per cell, since the input amount is known (100 FGOs vs 10000 mini oocytes).

Alternatively, the normalization could be based on the cellular mass (e.g. microgram of cell lysate).

As the data currently are presented, I have a concern that the protein intensity values of the mini oocytes may look higher than they actually are, because the Authors used 10000 mini oocytes vs 100 FGOs per sample. Considering the differences in cell size or volume (Fig.1E), I could accept that it takes 1000 (1 thousand) mini oocytes to generate the biomass of 100 FGOs, but the Authors used 10000 (10 thousand) mini oocytes - ten times more.

This could make a big difference to the interpretation of the data.

Please explain and justify how the proteomics data were normalized.

In Fig.5A and 6A for some proteins (IZUMO1R, NLRP9C, BMP15, GDF9, FIGLA, TBPL2, LHXB, DAZL, ZFP36L2) it is not clear if they were detected at very low level, or were not detected at all, in some samples. According to supplementary table S2, no-detection was the case. I suggest to write ND (not detected) in the white squares of the heat map, to make it clear that it is not a matter of low level, but of no detection.

Can the Authors please explain the apparent discrepancy between observing PADI6 in Fig.6A (proteome) and not observing cytoplasmic lattices in ultrastructure of the mini oocytes (Fig. S3). To my knowledge, PADI6 is integral component of the cytoplasmic lattices.

Line 586: "FGOs were collected by punching the ovaries of ICR female mice at 8-12 weeks of age"; perhaps the Authors would like to say 'poking' instead of 'punching'.

Line 257: "This analysis revealed the presence of a zona pellucida-like structure above the plasma membrane of mini-oocytes, although it was thinner and fainter than that of in vivo FGOs.". When the Authors say 'above' do they mean outside of the cell?

Figure 6D,F: did the knockout of Dppa3, Padi6, Khdc3, Nlrp5, Ooep and Tle6 reflect in reduced levels of these proteins, besides reduced level of the transcripts?

Title of Fig.5: it seems somewhat not representative of what is shown, because it mentions ultrastructure, but the figure shows also omics data and immunofluorescence.

Line 262: Authors note that cytoplasmic lattices were not detectable in the mini oocytes (Fig.S3A). I note that this could be a matter of stage of maturation (perhaps the cytoplasmic lattices are simply formed later in the mini oocytes).

Legend to Figure 2A: I suggest to mention also here which published datasets were retrieved from the literature and used for comparison (GSE183969, GSE79729, GSE128305).

Fig.2C and Fig.3B: why do the scatter plots look so different (more spreading of the data points in 2C than in 3B)? And why did the Authors use TPM in 2C vs RPM in 3B?

I find that the main points of the manuscript are adequately supported by the data presented, however I personally would tune down the statement that 'Mini-oocytes develop molecular networks observed in in vivo oocytes'. To be able to say so the Authors need to settle my previous critique about the normalization of the proteomics data. I suggest to say 'Mini-oocytes develop molecular networks reminiscent of in vivo oocytes'.

Overall, I enjoyed reading this manuscript and I think it has scientific merit. However, there are aspects that need to be clarified (first and foremost, the normalization behind the proteomics data of Fig.5A and 6A).

I think no additional experiments are required, although it would be very nice to show that spermatozoa can attach to the mini oocytes (this is expected, since the mini oocytes seem to produce an extracellular coat made of ZP2 and ZP3; whether the spermatozoa would also be able to penetrate the oolemma is difficult to tell, since IZUMO1 was not detected in the proteome of the mini oocytes).

I thank the Authors and the Journal for the opportunity to read this interesting study.

Point-by-point Responses to the Reviewers

Manuscript number: LSA-2025-03379-T

Revised title: A simple, efficient, and scalable method to generate oocyte-like cells *in vitro*

We appreciate the time and efforts that the reviewers put into reviewing our manuscript and find all comments helpful to improve our manuscript. According to their suggestions, we have revised our manuscript. Below you will find our point-by-point responses to the reviewers.

Reviewer #1 (Comments to the Authors (Required)):

The manuscript entitled "Establishment of a simple, efficient, and scalable method to generate oocyte like cells *in vitro*" by Banno et al. advances on the methodology for differentiating mouse pluripotent stem cells into cells of the female germline. The authors adapt a previously reported the transcription factor (TF)-based oocyte-like cell reprogramming method. In contrast, to the original protocol by Hamazaki et al. (2020) a 2D culture system without the need for gonadal somatic cells is used for generating so called "mini-oocytes". It is impressive to see that the resulting oocytes develop without the formation of typical follicle structure suggesting an largely cell autonomous process. Considering that oocytes are available in very limited numbers from mice, the approach is notable as a large number of oocytes can be obtained in relatively pure cultures (albeit data on homogeneity is missing) that greatly would facilitate proteomic, molecular analysis, and genomic screening. The authors perform a extensive characterization including morphology, size, transcriptome and proteome analysis, and molecular features of their mini-oocytes. These data suggest that minioocytes might resemble some characteristics of oocytes of primordial follicles. However, similarity is limited and it is clear that important differences exist that might limit the usefulness of minioocytes. In particular, meiosis is not represented as well as there is no opportunity to study the functionality of the ooplasm as the stage where activation can be considered is not reached, and hence embryonic potential (at least cleavage divisions) cannot be assessed. Although, the advance is encouraging the present version appears preliminary as no new mechanistic insight is reached and the suggested application for studying processes of oogenesis remains somewhat speculative. The impact of the study could be considerably improved by demonstrating a mechanistic advance beyond what has already been observed in *in vivo* oocytes. Yet, the presented characterization appears to remain on a level showing resemblance of processes in minioocytes to known regulation from *in vivo* oocytes.

Specific points

1. The advance over the original publication of TF mediated reprogramming and other systems that do not use gonadal somatic cells could be better described. The 8 TFs have been reported and a brief explanation of the inducer SHIELD would aid the reader. A focus on innovation in the culture beyond omitting

aggregation with gonadal somatic cells should be more clearly described. In particular, the culture relies on a number of previously suggested factors, and it is not clear what is new.

Response:

To clearly describe the advantage of our method, we have revised the text in Abstract and Results. Line 40-42: "This method requires minimal labor, does not rely on supporting somatic cells, and leverages a transcription factor-mediated approach for oocyte-like cell generation." Line 157-160: "Thus, this minioocyte induction system offers several advantages, including a simple procedure, no requirement for gonadal somatic cells, and an almost unlimited supply of oocyte-like cells, in contrast to previous oocyte induction systems." We also included additional explanation for Shield1. Line 122-125: "In the Shield-1 system, transcription factors are fused to a destabilization domain and actively degraded in the absence of Shield-1, while the addition of Shield-1 stabilizes these fusion proteins thereby enabling immediate and sustained transcription factor expression." Regarding the culture condition, we have revised the text as follows. Line 125-127: "For induction, a StemPro34-based medium containing Shield-1, stem cell factor (SCF), and Y-27632 (a ROCK inhibitor) was used, as this condition efficiently supported oocyte growth before (Hamazaki et al., 2021)."

Also it would be important to disclose why StemPRO has been used as for this culture medium it is unclear if the composition is known or commercially kept undisclosed. If the formulation would indeed be undisclosed this would lessen the enthusiasm considerably as biological understanding is likely limited.

Response:

We selected StemPro because it has previously been shown to support in vitro oocyte growth (Hikabe et al., 2016; Hamazaki et al., 2021), and in this study we prioritized its demonstrated effectiveness rather than the disclosed status of its composition. We agree with the reviewer's concern that the lack of compositional transparency is a limitation, and we acknowledge that future studies will benefit from either the disclosure of StemPro's formulation or the adoption of alternative media with defined compositions.

The authors should also consider the positioning of their minioocyte culture against earlier work of extended PGCLC culture that allows meiotic entry (Ohta et al. 2017, PMID: 28559416, Miyauchi et al., 2018).

Response:

As a paper reporting the generation of oocyte-like cells in vitro from mESCs without ovarian somatic cells (Nosaka et al., PMID: 40592343) was published while our manuscript was under revision, we have added a discussion of this study in Discussion (Line 419-423).

2. The authors perform extensive characterization of their minioocytes and use these data to illustrate similarities to in vivo oocytes. However, a number of important processes of oogenesis are not normal or represented in minioocytes including meiosis, formation of the zona pellucida, diameter remains below 40

micrometer. The lack of indication of functionality of the obtained minioocytes (activation, cleavage divisions) raise the concern that these might be poor approximations for many key processes and hence the question of applicability for future screening. A focus on presumably normal processes in minioocytes could help to understand future applications. A discussion of STRA8, ZGLP1 and FOXO3 expression data could also be useful for providing assessment. If an application for gaining novel mechanistic insight into one of these processes (Padi6 and STELLA regulation of UHRF1 are in the right direction but do not seem provide new insight against what is already known from in vivo oocytes) could be included, the study would more convincingly show that minioocytes are a useful model.

Response:

We appreciate the reviewer's insightful comments and fully agree with these points. To further assess the mini-oocyte induction process, we examined the expression of marker genes for primordial germ cells, meiosis, and oogenesis, including *Stra8*, *Zglp1*, and *Foxo3* (**new Fig. 2C and S2A**). This analysis showed that genes associated with oogenesis were strongly expressed during mini-oocyte induction, whereas markers for primordial germ cells and meiosis, such as *Prdm1*, *Stra8*, and *Spo11*, were not induced. As an additional response to the reviewer, we also examined FOXO3 localization in the presence or absence of SCF and found that FOXO3 is excluded from the nucleus in a SCF-dependent manner. This finding further suggested that in vivo oogenesis-like process occurs during mini-oocyte induction (**new Fig. 1I**).

Regarding the functionality of the obtained mini-oocytes, we previously attempted a sperm fusion assay to assess their ability to be fertilized. While we observed sperm attachment to mini-oocytes, sperm fusion was not detected. We did not include these data in the manuscript because we could not draw a clear conclusion: First, sperm attachment can also be seen when sperm are mixed with non-oocyte cells. Second, the sperm fusion assay requires removal of the zona pellucida, but it was difficult to confirm its digestion under a conventional optical microscope when using mini-oocytes. We therefore consider that these points need to be fully addressed in future studies.

We acknowledge that the current study does not yet provide novel mechanistic insights into oogenesis and agree that this represents an important future direction. Therefore, we plan to pursue it in follow-up studies.

3. Meiosis is not initiated in minioocytes but these cells are still said to arrest. As this is considered an important process in oogenesis, it would be good to discuss mitotic arrest in minioocytes in relation to meiotic prophase arrest. Also, migratory PGCs undergo transcriptional arrest and (later also dormancy) - would these processes be observed similar in minioocytes?

Response:

We discussed this point in the revised manuscript as follows: "Thus, TF-mediated activation of oocyte transcription network is sufficient to halt cell proliferation, whereas it remains unclear how cell cycle arrest occurs in the absence of meiosis" (line 167-169).

4. One remarkable aspect of the culture system is that apparently a large fraction of surviving cells

differentiate into minioocytes. However, this is stated in the text without any data about purity of the cultures cited. Could a more exact assessment be included what the percentage of oocytes and what other cell types (mesodermal?) in these cultures is?

Response:

We appreciate the reviewer's comment regarding the purity of the cultures. We investigated this point by assessing expression of oocyte growth markers, ZP2 and NPM2-mCherry reporter. This quantification indicated that 98.5 % of cells derived from 4F-mESCs were positive for ZP2, and 92.2 % of cells derived from 8F-mESCs were positive for NPM2-mCherry at day 21 (**new Fig. 1F and S1B**). These results demonstrate the high efficiency of mini-oocyte induction. Although we did not further characterize the rare marker-negative cells in the present study, we acknowledge that their identification will be important in our future work.

Minor comments

- The protocol adjustments, e.g. media changes, Shield1 timing, are useful but do not constitute a fundamentally new method. The title should be revised to avoid overstating novelty and instead emphasize practical refinements for scalability or screening. The abstract and introduction should also be adjusted accordingly.

Response:

We appreciate the reviewer's comment regarding the potential overstatement of novelty in our title. In response, we have revised the title to "A simple, efficient, and scalable method to generate oocyte-like cells in vitro", which we believe more accurately reflects the practical refinements and applicability of our study without overstating its novelty. We have also adjusted the abstract and introduction accordingly.

- P4, line 121 126 - Acknowledged that Shield1 as an inducer for overexpression of the PPT8 as described in Hamazaki et al., 2020, but maybe good to briefly explain the rationale.

Response:

We added the following sentences (line 122-125): "In the Shield-1 system, transcription factors are fused to a destabilization domain and actively degraded in the absence of Shield-1, while the addition of Shield-1 stabilizes these fusion proteins thereby enabling immediate and sustained transcription factor expression." The importance of TFs (PPT8) are explained at line 70-80.

- P5 - "Requirement of culture supplements for the generation of mini-oocytes" The culture conditions are also completely adapted from Hamazaki et al., 2020, except for the use of 10 μ M Y-27632 (vs. 10 nM in the original study). Was this concentration optimized here, or is it a discrepancy? Clarification is needed.

Response:

We appreciate the reviewer's careful reading. We confirmed that 10 μ M Y-27632 was used In Hamazaki et al., 2020 by contacting with the corresponding authors of that study, who are co-authors of the present manuscript. Although the culture condition was eventually almost identical to the one in Hamazaki et al.

2020, we believe that the data adjusting each concentration, which has not been shown before, is important.

- P6 - The authors note that 4F mini-oocytes adopt a round morphology by day 7, suggesting a critical transition stage. The expression of early germline markers, e.g., MVH, DAZL, should be demonstrated at this stage. In fact, key timepoints (e.g., days 7 and 21) should be tracked to validate progression.

Response:

In response to this suggestion and the comments above, we now show the expression of marker genes for primordial germ cells, meiosis, and oogenesis during mini-oocyte induction (**new Fig. 2C and S2A**). This analysis showed an upregulation of growing oocyte markers, including Zp1, Zp2, Zp3, and Gdf9. In contrast, upregulation of Dazl, an early germ cell marker, was not observed, possibly due to the absence of PGC formation in the mini-oocyte induction.

- P7 - The proteomic analysis and detection of ZP2/ZP3 (late-stage markers) are strengths. Are other key markers like NPM2 also expressed? And maybe point out in the main text as well if other key growth markers also expressing.

Response:

We appreciate the reviewer's suggestion. We now indicate the Npm2-mCherry reporter expression during mini-oocyte induction (new Fig. 1F and S1B). In addition, we show the expression of NPM2 as well as other late-stage markers, such as SCMC components, in Fig. 6A. We believe these additions appropriately address the reviewer's comment.

- Nobox is one of the inducible TFs the authors use for reprogramming. It should be clearly indicated that RNAseq and proteomics refers to the endogenous Nobox and not the transgenic factor.

Response:

We apologize for the lack of explanation. We included additional information in the corresponding figure legends as follows: Fig. 6(A) "Note that the expression of Figla, Tbp12, Nobox, and Lhx8 includes both endogenous and transgene expression".

- P25 - Figure 4, Proteome of mini-oocytes, at day 21 of culturing?

Response:

We apologize for the lack of explanation. We included additional information in the corresponding figure legends as follows: "(A) Venn diagrams showing the overlap of expressed genes. Genes detected by proteome analysis of *in vivo* FGOs, day 21 4F-mini-oocytes, and day 21 8F-mini-oocytes were compared."

- P7, line 244 - Could the authors briefly speculate on why ribosomal biogenesis proteins are overexpressed in mini-oocytes, for example, is this a compensatory stress response, culture artifact, or arrested developmental state?

Response:

We added the following discussion (line 270-272): "We speculate that these difference in proteome profiles between mini-oocytes and in vivo oocytes may reflect the culture environment, which substantially differs from its in vivo counterpart."

- P8, line 251- The title can be more precise, consider "Mini-Oocytes Partially Recapitulate the Ultrastructure of in vivo Oocytes"

Response:

We appreciate this kind suggestion. We changed the subheading considering the comments from two reviewers.

- P8 - The authors note that cytoplasmic lattices are absent in mini-oocytes. Did proteomic analysis reveal whether related proteins are expressed or if required signaling pathways are missing?

Response:

We observed that SCMC components required for cytoplasmic lattice are expressed. Recent studies indicate that factors beyond the canonical SCMC components, such as UHRF1, are also required for lattice formation (PMID: 37225425). We therefore speculate that mini-oocytes may lack, or insufficiently express additional factors that are essential for assembling cytoplasmic lattices. Further investigation is necessary in future to clarify this point. In addition, there is a remaining possibility that cytoplasmic lattices are formed in a small subset of mini-oocytes. Therefore, we revised the text and described a remaining possibility as follows (line 290-293): "In contrast, cytoplasmic lattices, a unique structure found in oocytes (Wassarman and Josefowicz, 1978), were not detectable in mini-oocytes, although we cannot exclude their presence in a subset of cells (Fig. S4A)."

- P8 - Validation of DDP3 KO efficiency is missing

Response:

We provide the data showing Dppa3 KO in **new Fig. S5A**.

- P8 - Does DDP3 KO only affect UHRF1 localization and not expression? unlike PADI6 loss?

Response:

In the DPPA3 KO experiments, we focused on the DPPA3-dependent UHRF1 cytoplasmic localization, which has been well demonstrated in in vivo oocytes (Li et al., 2018). Therefore, we examined UHRF1 localization rather than its expression.

- P9 - PADI6 loss affects UHRF1 expression and localization, maybe it would be useful to assess DNA methylation at a few loci e.g. imprinted genes as well? And similarly, for DDP3.

Response:

We appreciate the reviewer's insightful suggestion. The characterization of DNA methylation status,

particularly at imprinted loci, as well as other epigenetic signatures, is indeed highly relevant. While these analyses are beyond the scope of the present manuscript, we fully agree that they represent important future directions, and we plan to address them in follow-up studies.

- P9 - Is UHRF1 localization known to be affected by loss of other SCMC members in vivo?

Response:

We appreciate the reviewer's thoughtful comment. It is not known. Therefore, we revised the text and described this point as follows (line 349-352): "These findings align with a recent study that identified UHRF1 relocalization and reduction in Padi6 mutant mouse oocytes (Giaccari et al., 2024). On the other hand, whether the absence of other SCMC members affects UHRF1 localization in in vivo oocytes remains to be investigated."

- Since SCMC gene loss in mice causes multi-locus imprinting disturbances (MLID), it would be good to check whether other SCMC factors affect DNA methylation in mini-oocytes, even if UHRF1 localization is unchanged

Response:

We appreciate the reviewer's thoughtful suggestion. The characterization of DNA methylation and other epigenetic signatures is beyond the scope of the present manuscript. However, we fully agree that this highlights important future directions, and we plan to address these points in follow-up studies.

- Do DDPA3 and PADI6 KO alter mini-oocyte growth or morphology?

Response:

We have not observed any growth or morphological changes. We added this information in the revised manuscript (line 316-317 and 340-341).

- The transcriptomic and proteomic data may reveal why mini-oocytes exhibit restricted size. Are there deficiencies in growth-promoting pathways? Highlighting these could guide future optimizations.

Response:

We additionally performed KEGG pathway analysis and included this result in **new Fig. S3C**. We believe that this data might be informative for future optimization of culture conditions.

- In discussion, maybe good to suggest exploring additional growth signals to improve mini-oocyte size and maturation in future work for culture conditions.

Response:

By conducting KEGG pathway analysis, we found that genes related to the cAMP signaling pathway, which is involved in primordial follicle formation as well as meiotic arrest of oocytes were downregulated (**new Fig. S3C**). Based on this observation, we have added a discussion on the potential involvement of cAMP signaling in improving the mini-oocyte induction system in future studies (line 412-415).

- Overall, sub-headings and figure legends should be more detailed and concise.

Response:

We appreciate the reviewer's thoughtful suggestion. We revised the sub-headings and figure legends.

Reviewer #2 (Comments to the Authors (Required)):

Given the technical complexities of differentiating pluripotent e.g. embryonic stem cells (ESC) into oocytes in vitro, the Authors aimed to provide a simplified protocol that allows to generate oocytes in approximately 3 weeks, via forced expression of transcription factors (TFs) in a two-dimensional culture method, in the mouse model. Specifically, the Authors used mouse ESC lines in which the expression of 4 TFs (Nobox, Figla, Tbp12, and Lhx8) or 8 TFs (the 4 TFs plus Stat3, Dynll1, Sub1, and Sohlh1) can be induced in the presence of Shield-1. After 3 weeks, the ESC-derived oocytes were small (approx. 40 microns, hence called 'mini oocytes') but their transcriptomic and proteomic profiles were quite similar to those of oocytes fully grown in vivo. As proof of principle that the mini oocytes can be useful in studies of e.g. oocyte biology, CRISPR Cas9 technology was applied to inactivate genes of interest (Dppa3, Padi6, Nlrp5, Tle6, Ooep, Khdc3).

I have the following remarks.

How do the Authors explain that the zona pellucida was assembled outside the mini oocytes, although these were not encased in follicles? The ZP proteins are known to traffic independently of one another inside oocytes (PMID 17047254), and then are exocytosed. However, in the cell culture system used by the Authors, there is nothing that prevents the ZP proteins from diffusing away after they have been released outside. What was that kept the ZP proteins from diffusing away and allowed them instead to polymerize around the mini oocytes? Indeed, other secreted proteins like GDF9 and BMP15 diffuse away and are detected at low or minimal level in mini oocytes (Fig. 6A). Could the Authors please also add ZP1-2 to the heat maps of Fig.6A?

Response:

We appreciate the reviewer's comments. To avoid redundancy, we presented the heatmaps of ZP1, ZP2, and ZP3 in Fig. 5A. If the placement of the data in Fig. 6A is critically necessary, we are happy to add ZP1 and ZP2 expression profiles to Fig. 6A. Regarding the formation of the zona pellucida, previous studies have shown that it can be assembled without surrounding somatic cells (PMID:19247970; PMID:40592343), suggesting that somatic cells are not essential for its basic formation. Although the exact mechanism in mini-oocytes remains unclear, we speculate that local accumulation of ZP proteins and their intrinsic polymerization ability may account for their deposition around mini-oocytes rather than diffusing away. Nevertheless, we agree with the reviewer that somatic cells may contribute to the formation of a

thicker zona pellucida.

The heat maps in Fig.4B, Fig.5A and 6A could be easier to interpret. Are the color-coded values referring to 100 FGOs and 10000 mini oocytes, or are they referring to single cells? How was the normalization performed? In the transcriptome analysis the values shown in the heat maps are normalized (TPM, transcripts per million), but how have the results in the proteome analysis (data-independent acquisition mode) been normalized? In my opinion the values should be presented per cell, since the input amount is known (100 FGOs vs 10000 mini oocytes). Alternatively, the normalization could be based on the cellular mass (e.g. microgram of cell lysate). As the data currently are presented, I have a concern that the protein intensity values of the mini oocytes may look higher than they actually are, because the Authors used 10000 mini oocytes vs 100 FGOs per sample. Considering the differences in cell size or volume (Fig.1E), I could accept that it takes 1000 (1 thousand) mini oocytes to generate the biomass of 100 FGOs, but the Authors used 10000 (10 thousand) mini oocytes - ten times more. This could make a big difference to the interpretation of the data. Please explain and justify how the proteomics data were normalized.

Response:

We appreciate the reviewer’s comment regarding the normalization method used in the proteome analysis and apologize for the insufficient description of the method. As described in the revised methods section, the data were analyzed using DIA-NN (PMID: 31768060), which by default applies MaxLFQ normalization (PMID: 24942700). MaxLFQ normalization assumes that overall proteome abundance remains largely unchanged between any two samples. It calculates pairwise intensity ratios of all peptides commonly detected between any two samples (e.g., 100 FGO rep1 vs. 4F mini-oocytes 10⁴ rep1) and determines a normalization factor based on pairwise intensity ratios of commonly detected peptides. We confirmed that this approach was appropriate here because (1) the distributions and medians of log₁₀ intensities were comparable across all samples, and (2) pairwise replicate comparisons showed strong correlations along the y = x line, including the comparison between mini-oocytes and FGOs (see **Fig. R1** below).

While a proper data normalization was confirmed, we agree with the reviewer that the apparent overrepresentation of proteins in mini-oocytes may be technical, reflecting the difference in input amounts. Indeed, FGO samples showed lower total MS signal intensities. Therefore, we have now clearly stated in the revised manuscript that this overrepresentation may stem from this input difference (line 264-267 and 794-795).

Fig. R1 Validation of the proteome data normalization method
Left, Boxplots showing the distributions and medians of log₁₀-transformed intensities.
Right, Representative scatter plots showing the log₁₀-transformed intensity correlation between FGO (rep1) and mini-oocyte sample (8Fs rep1).

In Fig.5A and 6A for some proteins (IZUMO1R, NLRP9C, BMP15, GDF9, FIGLA, TBPL2, LHXB, DAZL, ZFP36L2) it is not clear if they were detected at very low level, or were not detected at all, in some samples. According to supplementary table S2, no-detection was the case. I suggest to write ND (not detected) in the white squares of the heat map, to make it clear that it is not a matter of low level, but of no detection.

Response:

We appreciate this kind suggestion. Gray squares indicated N.D. (not detected). We revised Fig. 5A, Fig. 6A, and corresponding figure legends accordingly.

Can the Authors please explain the apparent discrepancy between observing PADI6 in Fig.6A (proteome) and not observing cytoplasmic lattices in ultrastructure of the mini oocytes (Fig. S3). To my knowledge, PADI6 is integral component of the cytoplasmic lattices.

Response:

Initially, we expected to detect cytoplasmic lattices in mini-oocytes, since core components of the SCMC, including PADI6, are expressed. However, ultrastructural analysis did not reveal cytoplasmic lattices. Recent studies indicate that factors beyond the canonical SCMC components, such as UHRF1, are also required for lattice formation (PMID: 37225425). We therefore speculate that mini-oocytes may lack, or insufficiently express additional factors that are essential for assembling cytoplasmic lattices. Further investigation is necessary in future to clarify this point.

Line 586: "FGOs were collected by punching the ovaries of ICR female mice at 8-12 weeks of age"; perhaps the Authors would like to say 'poking' instead of 'punching'.

Response:

We appreciate the reviewer's careful reading. We have corrected it.

Line 257: "This analysis revealed the presence of a zona pellucida-like structure above the plasma membrane of mini-oocytes, although it was thinner and fainter than that of in vivo FGOs.". When the Authors say 'above' do they mean outside of the cell?

Response:

We appreciate the reviewer's careful reading. We intended to describe that the zona pellucida-like structure was located outside the plasma membrane of mini-oocytes. We have therefore revised the wording from "above" to "surrounding".

Figure 6D,F: did the knockout of Dppa3, Padi6, Khdc3, Nlrp5, Ooep and Tle6 reflect in reduced levels of these proteins, besides reduced level of the transcripts?

Response:

We appreciate the reviewer's thoughtful comment. We have shown the reduced levels of transcripts but not the protein levels in the original manuscript. Thus, we examined the protein levels of PADI6 in control

and knockout samples as a representative. We confirmed that the downregulation of PADI6 protein by western blotting (**new Fig. 6F**). In addition, we also confirmed by western blotting that Padi6 knockout is associated with the downregulation of UHRF1 protein levels (**new Fig. 6F and S5E**)).

Title of Fig.5: it seems somewhat not representative of what is shown, because it mentions ultrastructure, but the figure shows also omics data and immunofluorescence.

Response:

We agree with the reviewer, and therefore, we have changed the corresponding subheading and figure title to “Zona pellucida-like structures are formed in mini-oocytes”.

Line 262: Authors note that cytoplasmic lattices were not detectable in the mini oocytes (Fig.S3A). I note that this could be a matter of stage of maturation (perhaps the cytoplasmic lattices are simply formed later in the mini oocytes).

Response:

We appreciate the reviewer’s thoughtful comment. A previous study indicated that cytoplasmic lattices are already formed in immature growing oocytes (PMID: 18599511). Therefore, we consider it unlikely that the absence of cytoplasmic lattices in mini-oocytes is simply a matter of their maturation stage. Instead, we speculate that additional oocyte-specific factors, beyond the canonical SCMC components, may be required for their assembly, which will need to be investigated in future studies.

Legend to Figure 2A: I suggest to mention also here which published datasets were retrieved from the literature and used for comparison (GSE183969, GSE79729, GSE128305).

Response:

We added the information of the datasets in the legend of Fig. 2A.

Fig.2C and Fig.3B: why do the scatter plots look so different (more spreading of the data points in 2C than in 3B)? And why did the Authors use TPM in 2C vs RPM in 3B?

Response:

We show gene expression profiles in Fig. 2D (previously 2C), whereas Fig. 3B presents the expression of repetitive elements. TPM (transcripts per million mapped reads) was used in Fig. 2D to represent gene-length-normalized expression levels, as gene lengths vary substantially. In contrast, because the lengths of repetitive elements are relatively uniform, we used RPM (reads per million), which we believe is a common practice in the field. The broader distribution of data points in Fig. 2D compared to Fig. 3B is likely due to the fact that gene expression analysis deals with single-copy genes, where read counts vary more between replicates than in analyses of repetitive elements.

I find that the main points of the manuscript are adequately supported by the data presented, however I personally would tune down the statement that 'Mini-oocytes develop molecular networks observed in in

vivo oocytes'. To be able to say so the Authors need to settle my previous critique about the normalization of the proteomics data. I suggest to say 'Mini-oocytes develop molecular networks reminiscent of in vivo oocytes'.

Response:

We agree with the reviewer's comments, and therefore we changed the corresponding subheading to "Mini-oocytes develop molecular networks reminiscent of in vivo oocytes" (line 299).

Overall, I enjoyed reading this manuscript and I think it has scientific merit. However, there are aspects that need to be clarified (first and foremost, the normalization behind the proteomics data of Fig.5A and 6A).

Response:

We sincerely appreciate the reviewer's positive evaluation and constructive comments. We have revised the manuscript accordingly, and we hope that the current version fully addresses these concerns.

I think no additional experiments are required, although it would be very nice to show that spermatozoa can attach to the mini oocytes (this is expected, since the mini oocytes seem to produce an extracellular coat made of ZP2 and ZP3; whether the spermatozoa would also be able to penetrate the oolemma is difficult to tell, since IZUMO1 was not detected in the proteome of the mini oocytes).

Response:

We thank the reviewer for the comment that no additional experiments are required, and we agree that it is important to evaluate the mini-oocytes from different aspects. We previously attempted a sperm fusion assay to assess their ability to be fertilized. While we observed sperm attachment to mini-oocytes, sperm fusion was not detected. We did not include these data in the manuscript because we could not draw a clear conclusion: First, sperm attachment can also be seen when sperm are mixed with non-oocyte cells. Second, the sperm fusion assay requires removal of the zona pellucida, but it was difficult to confirm its digestion under a conventional optical microscope when using mini-oocytes. We therefore consider that these points need to be fully addressed in future studies.

I thank the Authors and the Journal for the opportunity to read this interesting study.

Response:

We sincerely appreciate the reviewer's positive evaluation and constructive comments.

November 4, 2025

RE: Life Science Alliance Manuscript #LSA-2025-03379-TR

Dr. Takashi Ishiuchi
University of Yamanashi
Faculty of Life and Environmental Sciences
4-4-37 Takeda
Kofu, Yamanashi 400-8510
Japan

Dear Dr. Ishiuchi,

Thank you for submitting your revised manuscript entitled "A simple, efficient, and scalable method to generate oocyte-like cells in vitro". Your revised manuscript was sent back to the original reviewers whose comments are appended below. As you will note, the reviewers are consistent in their evaluation that the revised study has addressed their concerns. Both reviewers had some follow-up points that we urge you to address in the manuscript text.

We would be happy to publish your paper in Life Science Alliance pending resolution of the reviewers' requests (above) and final changes necessary to meet our formatting guidelines.

- Please be consistent in describing numbers throughout the text (either spelt out or written as numbers, For example in line 719: "1,000 and ten mini-oocytes at day 7 and 21, respectively...")
- Please be consistent in use of italics font-type with 'in vivo' or 'in vitro' throughout the text.
- Kindly include the objective name, magnification, and NA whilst describing imaging in the methods section.
- Please explicitly state the availability of source data for the manuscript in the 'Data Availability' statement.
- Please upload your main manuscript text as an editable doc file.
- Please upload all figure files as individual ones, including the supplementary figure files.
- It is recommended to exclude figures from the manuscript text and upload them only separately.
- Please add a Summary Blurb/Alternate Abstract in our system.
- Please add the X and Bluesky handles of your host institute/organization as well as your own or/and one of the authors in our system.
- Please consult our manuscript preparation guidelines <https://www.life-science-alliance.org/manuscript-prep> and make sure your manuscript sections are in the correct order.
- Please use the [10 author names, et al.] format in your references (i.e., limit the author names to the first 10).
- Please add your main, supplementary figure, and table legends to the main manuscript text after the references section.
- Figure S1 and S4 have only one panel; therefore, please remove the label A from the current figures, their legends, and call-outs in the manuscript text.
- Please be sure that the authorship listing and order is correct.

LSA now encourages authors to provide a 30-60 second video where the study is briefly explained. We will use these videos on social media to promote the published paper and the presenting author (for examples, see <https://docs.google.com/document/d/1-UWCfbE4pGcDdcgzcmiuJl2XMBJnxKYeqRvLLrLS08s/edit?usp=sharing>). Corresponding or first-authors are welcome to submit the video. Please submit only one video per manuscript. The video can be emailed to contact@life-science-alliance.org

A. FINAL FILES:

- An editable version of the final text (.DOC or .DOCX) is needed for copyediting (no PDFs).
- High-resolution figure, supplementary figure and video files uploaded as individual files: See our detailed guidelines for

preparing your production-ready images, <https://www.life-science-alliance.org/authors>

B. MANUSCRIPT ORGANIZATION AND FORMATTING:

Thank you for your attention to these final processing requirements. Please revise and format the manuscript and upload materials as soon as you are able.

Sincerely,

Sarita Hebbar, PhD
Scientific Editor
Life Science Alliance
<http://www.lsajournal.org>

Reviewer #1 (Comments to the Authors (Required)):

The revised version of the manuscript entitle "A simple, efficient, and scalable method to generate oocyte-like cells in vitro" by Banno et al. provides a well positioned and detailed report of the development of a new culture system that allows to obtain oocyte like cells from pluripotent mouse stem cell cultures with resemblance of fully grown oocytes of mice within 3 weeks. In their revision the authors have added additional data and made changes to the text that have further strengthened the study. In particular the cell composition and homogeneity has been investigated by two markers (ZP2 and NPM2-reproter). These data show that the cultures contain over 90 percent cells specified as oocytes on day 21, which is impressive and will be enabling for future experimental strategies. Culture media optimization, DNA methylation or chromatin analyses, genetics and biochemical approaches will undoubtedly extended beyond prior limitations. These advantages are somewhat offset by the imperfect approximation of embryonic oocytes. I do agree with the authors notion that the absence of meiosis in this very specific TF induced differentiation/reprograming system might not to be a disadvantage but a potential strength, as it conceptually allows to separate meiosis from the oogenic differentiation program, which has potential for unravelling the intricate molecular regulatory (TF) networks in future work. However, the revised version has improved the positioning of the study and can convincingly demonstrate that a large advance has been made.

The authors illustrate the use of their new culture system by performing a small scale genetic investigation of components of the subcortical maternal complex which confirms previous results in oocytes and also makes interesting observations. The additional data and extensive changes to the text have further strengthened the study, which can be expected to meet high interest of a wide readership audience of researchers working on stem cells, germline development, and reproductive biology. The revised version has addressed my earlier concerns and I recommend considering publication of the revised study.

Minor point

For the methods section a small clarification on page 20 section starting at line 674 briefly stating that the 4F ESCs were infected with Cas9 lentivirus and selected with blasticidin and following a resistant polyclonal pool was used for transfection with the zeozin resistance conferring sgRNA plasmid (U6.. or is this also a lentivirus) could help to make this experiment easier to understand fully.

Reviewer #2 (Comments to the Authors (Required)):

I have reviewed both, the initial and also the revised version of the manuscript titled "A simple, efficient, and scalable method to generate oocyte-like cells in vitro." Therefore, I omit here a short summary of the manuscript, since I provided it before. My overall assessment of the work has not changed and is positive.

The Authors produced a thorough revised version. Above all, I appreciate that the possibility of different input amounts leading to apparent higher abundance of certain proteins in mini-oocytes has now been recognized and mentioned explicitly in the revised text. I also appreciate the positive actions that have been taken in response to my other critiques. The Authors identified with ND the proteins that were not detected in the mini oocytes. The information about the source datasets has now been added to the legend of Fig. 2A. The Authors adopted my suggestion and modified the title of a section ("Mini-oocytes develop molecular networks reminiscent of in vivo oocytes"). The difference between the scatter plots in Fig.2C and Fig.3B has been explained in the rebuttal. Overall, the rebuttal has answered my questions in a satisfactory manner.

I still have just one strong suggestion and a couple of minor things I would like clarification on.

Strong suggestion: the current title is "A simple, efficient, and scalable method to generate oocyte-like cells in vitro." Well, from all the things that have been written in the manuscript and said during the review process, I understand that the mini oocytes are 'similar' to the oocytes of primary follicles, which have a diameter of 25-30 microns. Therefore, I suggest that this state should be mentioned in the title, for example: "A simple, efficient, and scalable method to generate oocyte-like cells similar to those of primordial follicles in vitro". Otherwise, readers may think the products of the study attained full-size, while in fact they were mini. I think this distinction should be communicated in the title, so as not to raise too high expectations.

The mini-oocytes are positive to PADI6 yet the cytoplasmic lattices are not detectable in ultrastructure. The Authors suggest that this is because the lattices are composed of many other things besides PADI6, therefore, mini-oocytes do not form lattices because PADI6 is present but other components are still missing. This is a reasonable possibility. I would like to insist on the size factor though. Mini oocytes are up to 40 microns in diameter. To my knowledge, proper cytoplasmic lattices form at the end of oogenesis in mice, see Yurttas et al., Development 2008 PMID 18599511 ("CPLs first become evident in Padi6^{+/+} oocytes at approximately the 40 μ m stage of growth and appear to be derived from curvilinear 'intermediate structures' (IS), which have properties of both aggregated ribosomes (Rb) and mature lattices"). I personally would not expect lattices to be found in mouse oocytes at 40 microns diameter or less. In think size is the simpler explanation for the lack of mature cytoplasmic lattices in mini oocytes.

The staining of the zona pellucida Fig.5C,D is convincing, because the antibodies used are the best (ZP2 and ZP3 hybridoma antibodies of unquestionable specificity). Yet I find it surprising that the immunostaining for ZP2 and ZP3 does not detect intracellular ZP proteins. Before being secreted, ZP proteins should be found in the cytoplasm (see Fig.1 in Hoodbhoy et al., Mol Cell Biol. 2006 PMID 17047254), while Figs.5C,D are dark in the space between nucleus and cell membrane. How do the Authors explain the discrepancy? Could the Authors also please clarify if the FGO of Fig.5C,D had the zona pellucida removed with acidic tyrode solution or other method prior to immunofluorescence. What I am trying to understand is whether the staining in Fig.5C,D pertains to the secreted, extracellular zona or to the zona proteins still attached to the oolemma.

I thank the Authors for the thorough revision, which meets my satisfaction. These final comments are minor and should not hold the manuscript from a positive decision.

Point-by-point Responses to the Reviewers

Manuscript number: LSA-2025-03379-TR

Title: A simple, efficient, and scalable method to generate oocyte-like cells *in vitro*

We appreciate the time and efforts that the reviewers put into reviewing our manuscript and find all comments helpful to improve our manuscript. According to their suggestions, we have revised our manuscript. Below you will find our point-by-point responses to the reviewers.

Reviewer #1 (Comments to the Authors (Required)):

The revised version of the manuscript entitled "A simple, efficient, and scalable method to generate oocyte-like cells *in vitro*" by Banno et al. provides a well positioned and detailed report of the development of a new culture system that allows to obtain oocyte like cells from pluripotent mouse stem cell cultures with resemblance of fully grown oocytes of mice within 3 weeks.

In their revision the authors have added additional data and made changes to the text that have further strengthened the study. In particular the cell composition and homogeneity has been investigated by two markers (ZP2 and NPM2-reproter). These data show that the cultures contain over 90 percent cells specified as oocytes on day 21, which is impressive and will be enabling for future experimental strategies. Culture media optimization, DNA methylation or chromatin analyses, genetics and biochemical approaches will undoubtedly extended beyond prior limitations. These advantages are somewhat offset by the imperfect approximation of embryonic oocytes. I do agree with the authors notion that the absence of meiosis in this very specific TF induced differentiation/reprogramming system might not to be a disadvantage but a potential strength, as it conceptually allows to separate meiosis from the oogenic differentiation program, which has potential for unravelling the intricate molecular regulatory (TF) networks in future work. However, the revised version has improved the positioning of the study and can convincingly demonstrate that a large advance has been made.

The authors illustrate the use of their new culture system by performing a small scale genetic investigation of components of the subcortical maternal complex which confirms previous results in oocytes and also makes interesting observations.

The additional data and extensive changes to the text have further strengthened the study, which can be expected to meet high interest of a wide readership audience of researchers working on stem cells, germline development, and reproductive biology. The revised version has addressed my earlier concerns and I recommend considering publication of the revised study.

Response:

We deeply appreciate the reviewer's fair and thorough evaluation of our study.

Minor point

For the methods section a small clarification on page 20 section starting at line 674 briefly stating that the 4F ESCs were infected with Cas9 lentivirus and selected with blasticidin and following a resistant polyclonal pool was used for transfection with the zeozin resistance conferring sgRNA plasmid (U6.. or is this also a lentivirus) could help to make this experiment easier to understand fully.

Response:

We apologize that our description of this method section was not sufficient. According to the suggestion from the reviewer, we have added the text as follows: "In brief, 4F-mESCs were infected with Cas9 lentivirus and selected with blasticidin, and a single blasticidin-resistant clone efficiently expresses Cas9 was used for transfection. For sgRNA expression, oligos containing sgRNA target sequences were first inserted into a constructed sgRNA expressing vector that confers zeocin-resistance (U6-sgRNA-CAG-Zeo)".

Reviewer #2 (Comments to the Authors (Required)):

I have reviewed both, the initial and also the revised version of the manuscript titled "A simple, efficient, and scalable method to generate oocyte-like cells in vitro." Therefore, I omit here a short summary of the manuscript, since I provided it before. My overall assessment of the work has not changed and is positive. The Authors produced a thorough revised version. Above all, I appreciate that the possibility of different input amounts leading to apparent higher abundance of certain proteins in mini-oocytes has now been recognized and mentioned explicitly in the revised text. I also appreciate the positive actions that have been taken in response to my other critiques. The Authors identified with ND the proteins that were not detected in the mini oocytes. The information about the source datasets has now been added to the legend of Fig. 2A. The Authors adopted my suggestion and modified the title of a section ("Mini-oocytes develop molecular networks reminiscent of in vivo oocytes"). The difference between the scatter plots in Fig.2C and Fig.3B has been explained in the rebuttal. Overall, the rebuttal has answered my questions in a satisfactory manner.

Response:

We deeply appreciate the reviewer's fair and thorough evaluation of our study.

I still have just one strong suggestion and a couple of minor things I would like clarification on.

Strong suggestion: the current title is "A simple, efficient, and scalable method to generate oocyte-like cells in vitro." Well, from all the things that have been written in the manuscript and said during the review process, I understand that the mini oocytes are 'similar' to the oocytes of primary follicles, which have a diameter of 25-30 microns. Therefore, I suggest that this state should be mentioned in the title, for example: "A simple, efficient, and scalable method to generate oocyte-like cells similar to those of primordial follicles in vitro". Otherwise, readers may think the products of the study attained full-size, while in fact they were

mini. I think this distinction should be communicated in the title, so as not to raise too high expectations.

Response:

We appreciate the reviewer's thoughtful suggestion. However, we feel that the suggested title would not accurately represent our study, as our transcriptome analysis showed that mini-oocytes generated after 21 days of culture are more similar to fully grown oocytes (FGOs) than to oocytes of primordial follicles. To clarify that our oocyte-like cells are smaller than FGOs, we refer to them as mini-oocytes in the Abstract and throughout the manuscript. In line with the reviewer's advice, we also revised a sentence in the Abstract to read: "Our transcriptome and proteome analyses revealed significant similarities between in vitro-derived mini-oocytes and in vivo oocytes, despite their relatively small size." We believe that this clarification in the Abstract appropriately informs readers of the size difference.

The mini-oocytes are positive to PADI6 yet the cytoplasmic lattices are not detectable in ultrastructure. The Authors suggest that this is because the lattices are composed of many other things besides PADI6, therefore, mini-oocytes do not form lattices because PADI6 is present but other components are still missing. This is a reasonable possibility. I would like to insist on the size factor though. Mini oocytes are up to 40 microns in diameter. To my knowledge, proper cytoplasmic lattices form at the end of oogenesis in mice, see Yurttas et al., Development 2008 PMID 18599511 ("CPLs first become evident in Padi6+/- oocytes at approximately the 40 μ m stage of growth and appear to be derived from curvilinear 'intermediate structures' (IS), which have properties of both aggregated ribosomes (Rb) and mature lattices"). I personally would not expect lattices to be found in mouse oocytes at 40 microns diameter or less. I think size is the simpler explanation for the lack of mature cytoplasmic lattices in mini oocytes.

Response:

We would like to thank the reviewer for the thoughtful comment. We agree that the smaller size of mini-oocytes can be a cause of the lack of cytoplasmic lattices. We therefore consider that these points need to be fully addressed in future studies by improving our method.

The staining of the zona pellucida Fig.5C,D is convincing, because the antibodies used are the best (ZP2 and ZP3 hybridoma antibodies of unquestionable specificity). Yet I find it surprising that the immunostaining for ZP2 and ZP3 does not detect intracellular ZP proteins. Before being secreted, ZP proteins should be found in the cytoplasm (see Fig.1 in Hoodbhoy et al., Mol Cell Biol. 2006 PMID 17047254), while Figs.5C,D are dark in the space between nucleus and cell membrane. How do the Authors explain the discrepancy? Could the Authors also please clarify if the FGO of Fig.5C,D had the zona pellucida removed with acidic tyrode solution or other method prior to immunofluorescence. What I am trying to understand is whether the staining in Fig.5C,D pertains to the secreted, extracellular zona or to the zona proteins still attached to the oolemma.

Response:

We thank the reviewer for the insightful comment. As noted, ZP proteins are synthesized in the cytoplasm before secretion. Indeed, we observed weak cytoplasmic ZP signals in mini-oocytes, which can be faintly

seen in Fig. 5C (ZP2 staining of mini-oocytes). However, since only a single optical plane from confocal microscopy is shown, these cytoplasmic signals are not clearly visible. Regarding the reviewer's question about zona pellucida removal, the zona pellucida was not removed prior to immunostaining. Therefore, the staining in Fig. 5C,D likely represents both the extracellular zona pellucida and ZP proteins attached to the oolemma, which cannot be clearly distinguished in this analysis. Nevertheless, based on our electron microscopy observations, we consider that the immunofluorescence signals at least include detection of extracellular ZP proteins.

I thank the Authors for the thorough revision, which meets my satisfaction. These final comments are minor and should not hold the manuscript from a positive decision.

Response:

We deeply appreciate the reviewer's fair and thorough evaluation of our study, which was very helpful in improving our manuscript.

November 7, 2025

RE: Life Science Alliance Manuscript #LSA-2025-03379-TRR

Dr. Takashi Ishiuchi
University of Yamanashi
Faculty of Life and Environmental Sciences
4-4-37 Takeda
Kofu, Yamanashi 400-8510
Japan

Dear Dr. Ishiuchi,

Thank you for submitting your Research Article entitled "A simple, efficient, and scalable method to generate oocyte-like cells in vitro". It is a pleasure to let you know that your manuscript is now accepted for publication in Life Science Alliance. Congratulations on this interesting work.

Your manuscript will now progress through copyediting and proofing. At the proofs stage you may include a clarification, in the manuscript text, about the staining in Fig 5C/D (in response to the reviewer's comments).

It is journal policy that authors provide original data upon request.

DISTRIBUTION OF MATERIALS:

Again, congratulations on a very nice paper. I hope you found the review process to be constructive and are pleased with how the manuscript was handled editorially. We look forward to future exciting submissions from your lab.

Sincerely,

Sarita Hebbar, PhD
Scientific Editor
Life Science Alliance
<http://www.lsajournal.org>